# Seasonal and Spatial Changes in Trace Gases over Megacities from AURA TES Observations: Two Case Studies

Karen E. Cady-Pereira[1], Vivienne H. Payne[2], Jessica L. Neu[2], Kevin W. Bowman[2], Kazuyuki Miyazaki[2,3], Eloise A. Marais[4*], Susan Kulawik[5], Zitely A. Tzompa-Sosa[6], Jennifer D. Hegarty[1]

[1]Atmospheric and Environmental Research, Lexington, MA, USA

[2]Jet Propulsion Laboratory, California Institute of Technology, Pasadena, CA, USA

[3]Japan Agency for Marine-Earth Science and Technology, Yokohoma, Kanagawa Prefecture, Japan

[4]School of Engineering and Applied Sciences, Harvard University, Cambridge, MA, USA

[5]Bay Area Environmental Research Institute/NASA Ames, Mountain View, CA, USA

[6]Department of Atmospheric Sciences, Colorado State University

* Now at: School of Geography, Earth and Environmental Sciences, University of Birmingham, Edgbaston, UK

*Correspondence to*: K. E. Cady-Pereira (kcadyper@aer.com)

## Abstract

The AURA TES instrument is collecting closely spaced observations over 19 megacities. The objective is to obtain measurements that will lead to better understanding of the processes affecting air quality in and around these cities, and better estimates of the seasonal and interannual variability. We explore the TES measurements of ozone, ammonia, methanol and formic acid collected around the Mexico City Metropolitan Area and in the vicinity of Lagos (Nigeria). The TES data exhibit seasonal signals that are correlated with AIRS CO and MODIS AOD, with in situ measurements in the MCMA and with GEOS-Chem model output in the Lagos area. TES was able to detect an extreme pollution event in the MCMA on April 9, 2013, which is also evident in the in situ data. TES data also show that biomass burning has a greater impact south of the city than in the caldera where Mexico City is located. TES measured enhanced values of the four species over the Gulf of Guinea south of Lagos. Since it observes many cities from the same platform, with the same instrument and applies the same retrieval algorithms, TES data provide a very useful tool for easily comparing air quality measures of two or more cities. We compare the data from the MCMA and Lagos, and show that while the MCMA has occasional extreme pollution events, Lagos consistently has higher levels of these trace gases.

## 1 Introduction

The world's megacities, defined as urban agglomerations with a population over ten million (UN, 2016), currently house close to 500 million people, about one eighth of the global urban population

(UN, 2014). They have increased in number, from two in 1950 (New York-Newark and Tokyo-Yokohama), to 28 in 2014, and have also become increasingly larger. This growth is expected to continue, as more and more people leave rural areas for better opportunities and a higher quality of life. Yet while it has been shown that urban dwellers are financially better off, better educated and better fed, this improvement does come at the cost of strained services and increased congestion, causing, among other problems, increases in urban water and air pollution. The fastest growing megacities in the past few decades have been located primarily in the developing world (UN, 2014), where rapid industrialization in combination with high population density and limited emission controls is leading to serious air quality problems (UN, 2014). Since air quality directly affects human health, contributing significantly to the incidence of respiratory and cardiovascular diseases (Pope et al, 2002; Pope et al., 2009), straining health care facilities and driving health costs up, many megacities are directing intense efforts to improve air quality by controlling emissions.

In addition to local sources, many cities must also contend with pollution transported from surrounding areas. Upwind industrial activities can severely impact air quality. In the tropics, smoke from biomass burning may often exacerbate existing air quality problems in population centers [Pavagadhi et al., 2013; Marais et al; 2014; Yokelson et al., 2011; Kreidenweis et al., 2001]. At low latitudes high UV levels increase $O_3$ formation rates, and high temperatures result in high levels of Volatile Organic Compounds (VOCs) emissions and fast photochemistry. In addition to the issues of air quality within megacities, urban/industrial emissions from these cities can change the chemical content of the downwind troposphere, influencing both air quality and climate on scales ranging from regional to continental and global (Molina and Molina, 2002; Molina et al., 2004; Parrish et al., 2009; Molina et al., 2010).

The impact of pollution on air quality and human health has led to considerable effort to measure trace gases, using surface networks, aircraft campaigns and satellite. Each approach has its own advantages and limitations. Surface monitoring networks provide in situ data on the air people actually breathe, but they are often sparse, especially in the developing world, and cannot provide much information on the vertical distribution of pollutants. Furthermore, some chemical species have high spatial variability that a sparse network cannot capture. Aircraft campaigns, such as the Megacity Initiative: Local and Global Research Observations, MILAGRO (Singh et al., 2009), the Intercontinental Transport Experiment-Phase B, INTEX-B (Singh et al., 2009) in Mexico, and the African Monsoon Multidisciplinary Analysis (AMMA) (2006) yield valuable information on the vertical distribution of pollutants and their precursors, provide contextual background for air quality events, and significantly add to our understanding of regional processes; however, campaigns are expensive, and their temporal coverage is very limited. Satellite data have coarser spatial resolution and less capability to provide near surface measurements but offer much greater temporal and spatial coverage than aircraft campaigns. They can fill the need for extensive data over long periods and large regions; their data can be used to determine

the larger scale spatial variability and the seasonal and interannual signals of atmospheric pollutants and provide useful constraints on model emissions and chemistry. Duncan et al. (2016) were able to calculate 10 year trends in $NO_2$ from the Ozone monitoring Instrument (OMI) over nearly 200 cities worldwide, and to examine how these trends varied spatially: for example they found that $NO_2$ increased north and west of the city but decreased in the center and to the northeast. Similarly Jiang et al. (2012) examined trends in total column $SO_2$ from OMI over China.

Here we will focus on data from the Tropospheric Emission Spectrometer (TES) flying on the NASA AURA satellite. TES data have been used extensively to evaluate and improve Chemical Transport Model (CTM) performance in previous air quality studies. TES ozone ($O_3$) data were assimilated into the Real-time Air Quality Modeling System (RAQMS) model and reduced model biases with respect to aircraft measurements during the Second Texas Air Quality Study (TexAQS II) (Pierce et al., 2009). Ammonia ($NH_3$) data from TES were used in an inverse modeling approach to improve emissions in the GEOS-Chem model, leading to better agreement with AMoN surface data (Zhu et al., 2013). Similarly, TES methanol ($CH_3OH$) data were also used to optimize GEOS-Chem methanol emission, leading to better agreement between the model and aircraft measurements (Wells et al., 2014), obtained from a number of campaigns, including MILAGRO (Singh et al., 2009), INTEX-B (Singh et al., 2009), the second Texas Air Quality Study, TexAQS-II (Parrish et al., 2009), and the Arctic Research of the Composition of the Troposphere from Aircraft and Satellites, ARCTAS (Jacob et al., 2010).

The studies above used measurements from two of the TES observing modes that provide near-global (Global Surveys) or extensive regional (Step and Stares) coverage. To reduce wear and tear associated with the aging of the instrument and extend its lifetime, a new limited observing strategy was initiated in January 2013. This strategy reduces the number of sample points globally, but increases their density over designated regions, including 19 selected megacities (https://tes.jpl.nasa.gov/visualization/SCIENCE_PLOTS/SO/main_Transect_megacity_locations.html).

This strategy implemented Special Observations (SOs) known as transects. Each TES megacity transect contains 20 footprints spaced 12 km apart with simultaneous measurements (at approximately 1:30 pm local time) of a number of important trace gases, which provide a chemical snapshot of each city every two weeks, and can be used to determine the trace gas distribution and variability. In this study we use the megacity data to show variations of four key trace gases related to air quality, selected from the much larger set available: $O_3$, $NH_3$, $CH_3OH$ and formic acid (HCOOH)). Ozone is a secondary pollutant formed by the reaction of $NO_x$ and VOCs (Crutzen et al., 1988, Jacob et al., 1996), and is harmful to human and ecosystem health. $CH_3OH$ is principally produced by plants (MacDonald and Fall, 1993). Oxidation of $CH_3OH$ forms carbon monoxide and formadelhyde, both of leading to the production of (Tie et al., 2003). The main source of $NH_3$ is agricultural activity (Bouwman et al., 1997, Paulot and Jacob, 2014) but it is also formed by catalaytic converters (Nowak et al., 2012), which can be a significant source in large metropolitan areas; $NH_3$ reacts with sulphuric acid and nitric acid to form

small aerosol particles. HCOOH is largely formed by photochemical oxidation of VOCs in the troposphere, but is also directly emitted by vegetation and other sources (Millet et al., 2015). $NH_3$, HCOOH (R'Honi et al., 2013) and $CH_3OH$ (Holzinger et al., 1999) are also by-products of biomass burning, and are correlated with CO during biomass burning events (Luo et al., 2015). While carbon monoxide (CO) observed by TES has provided extremely useful information as a tracer for combustion in the past (Luo et al., 2007; Worden et al.; 2013, Luo et al., 2015), current TES CO throughput is very poor (due to issues with the TES filter in the range sensitive to CO) and has not been included in this study.

We focus on two metropolitan areas, Mexico City, Mexico and Lagos, Nigeria. Both cities are located at low latitudes, but they lie in very different geographic contexts. Mexico City is landlocked at 2200 m altitude on the Tropic of Cancer, while Lagos lies at sea level on the Gulf of Guinea at six degrees north of the Equator., Both are also located in regions with strong seasonal biomass burning. We use in situ measurements (in Mexico City) and back trajectories to assess the influence of biomass burning on the air quality within the cities themselves, and we demonstrate the use of the TES observations to provide finer scale information beyond that derived from chemical transport models in these regions.

Section 2 briefly describes the TES instrument and the characteristics of the TES megacity observations, as well as ancillary measurements and models used. Section 3 presents an analysis of the TES data over Mexico City and Lagos, focusing on how emissions from surrounding areas, including biomass burning and industrial sources, play a role in determining local air quality. Section 4 summarizes our results.

## 2 Data and Models

### 2.1 TES

TES is a nadir-viewing Fourier-transform infrared (FTIR) spectrometer with a high spectral resolution (0.06 cm$^{-1}$) and a nadir footprint of 5.3 km x 8.3 km (Beer, 2006). TES flies aboard the NASA Aura spacecraft, which was launched in July 2004 and is part of the A-train constellation of satellites. It is in a sun-synchronous orbit with an equator crossing time at around 01:30 and 13:30 local solar time. While TES does not provide the broad spatial coverage offered by scanning instruments such as the Metop Infrared Atmospheric Sounding Interferometer (IASI) or the NASA Atmospheric Infrared Sounder (AIRS), its higher spectral resolution yields greater sensitivity to gas concentrations lower in the atmosphere (Clarisse et al., 2010). The standard mode of TES observations until 2011 was Global Surveys (GS), which consisted of widely spaced observations (~180 km apart) collected during 15 orbits (approximately 26 hours). During the gaps between GS observations, focused Special Observations (SOs) were carried out, with various different objectives including colocation with aircraft campaigns (e.g., MILAGRO) and detailed studies of regions with significant pollution, such as eastern

China or the Fort McMurray area in Alberta. SOs are more closely spaced (12 to 60 km), and extend over a few hundred to a thousand kilometres, providing more detailed information on regional variability. After 2011, SOs became the only operational mode. The TES Megacity observation strategy was implemented in January 2013 with the goal of providing data to improve our knowledge of the air

quality over regions where a large fraction of the world's population lives. Each SO run, or transect, consists of 20 closely spaced footprints (12 km apart), running almost north to south (Figure 1), repeated every 16 days. Nineteen metropolitan areas were chosen based on their population and known air quality problems. Due to satellite viewing limitations, the transects do not necessarily cross directly over the center of each city.

TES retrievals follows an optimal estimation approach (Rodgers, 2000; Bowman et al., 2006) that minimizes the differences between the TES Level 1B spectra and a radiative transfer calculation that uses absorption coefficients calculated with the AER line-by-line radiative transfer model LBLRTM (Clough et al., 2006). The optimal estimation algorithm produces the averaging kernels and error covariance matrices necessary for utilization of the data within data assimilation and inverse modelling

frameworks. While the TES sensitivity to $O_3$ peaks between 600 and 400 mbar (Worden et al., 2007), for the other trace gases discussed here the sensitivity peaks much lower in the atmosphere: between 900 and 700 mbar for $NH_3$ (Shephard et al., 2011) and $CH_3OH$ (Cady-Pereira et al., 2012), and between 900 and 600 mbar for $HCOOH$ (Cady-Pereira et al., 2014). The early afternoon overpass time is ideal for observing these gases, which tend to be mostly concentrated in the lower troposphere, as the thermal

contrast between the surface temperature and the temperature at the peak of the TES sensitivity is enhanced, providing stronger radiative signals.

One of the main objectives of this analysis was to understand the temporal and spatial variability of the trace gases. This analysis is simpler if a representative value is selected. A basic approach is to select a retrieval level (e.g., 918 mbar) and construct time series and spatial gradients at that level, as has been

done by Warner et al. (2016) in their analysis of AIRS $NH_3$. However, this approach disregards several important facts: first, the retrievals of $NH_3$, $CH_3OH$ and $HCOOH$ have limited information, on the order of 1.0 Degrees of Freedom for Signal (DOFS), which means that the a priori profile shape strongly influences the retrieved value at any level; second, the sensitivity at each level varies significantly from profile to profile. Thus the difference between the values from two retrievals at a

selected level may often be a difference in a priori or sensitivity.

In order to maximize the information from the retrieval and reduce the impact of the a priori Payne et al. (2009) adapted the mapping approach Von Clarmann and Grabowski (2007) developed for limb retrievals with relatively high DOFS to nadir retrievals with low DOFS, to generate a single representative value for each retrieved profile, containing all the available information. However, the

mapping approach can be numerically problematic for profiles with peak concentrations at the surface; in this work instead, the levels of $NH_3$, $CH_3OH$ and $HCOOH$ with highest sensitivity, as determined by

the diagonal of the averaging kernel, were averaged; usually these are the level of the peak of the averaging kernel and the levels immediately below, if the peak is not at the surface. Thus the sensitivity of the retrieval is used to combine the levels with the most information.

While these representative values are not direct surface measurements, our experience is the representative values for $NH_3$, $CH_3OH$ and $HCOOH$ will be well correlated with surface values; in other words, the spatial gradients and temporal variability of these values will be very similar to gradients and temporal signals of the surface measurements, (e.g. Pinder et al., 2011, Dammers et al., 2017). For these species most of the gas is concentrated in or just above the boundary layer, therefore TES is measuring concentrations close to the surface; furthermore, both $NH_3$ and $HCOOH$ are radiatvely active in spectral windows, and will dominate the TES signal in these regions. Moreover, since the TES cross-over time is at 1:30 pm local time, TES is observing at the time of day when the boundary layer tends to be thicker and more well mixed, and thus the TES observation is likely to be closer to the surface value.

Since the $O_3$ averaging kernel peaks at much greater altitudes, $O_3$ is distributed over entire troposphere and its concentration peaks in the stratosphere, the above approach would not provide information on near surface values. The DOFS for the first three levels were analyzed and were found to range between 0.2 and 0.5; thus the retrievals results at these levels have some sensitivity to the $O_3$ amounts in this region, and are not simply being driven by the a priori (see Appendix). Chatfield and Esswein (2012) have shown that $O_3$ partial columns over the first 3 km above the surface have correlations with surface $O_3$ ranging from 0.41 to 0.94 for a set of sondes stations across North America. This altitude range roughly corresponds to the first three TES levels. Based on these two observations we have chosen the average of these first three levels as a representative value for $O_{3.}$.

Only profiles with good quality flags are included in this processing. The representative values are averaged over each transect to generate time series, and seasonally over each observation point to produce plots of spatial variability along each transect. The TES Megacity data are TES Version V006 and are available at

https://avdc.gsfc.nasa.gov/index.php?site=635564035&id=10&go=list&path=/Megacity.

## 2.2 Other satellite data

Time series of CO V006 data from the NASA AIRS instrument, along with aerosol optical depth (AOD) at 550 nm from the NASA Moderate Resolution Imaging Spectroradiometer (MODIS) Deep Blue Algorithm are presented to provide context for the TES data. These are daily gridded products on a 1° by 1° grid, obtained from the NASA Giovanni site (http://giovanni.gsfc.nasa.gov/giovanni/), and have been averaged over the length of the transect. Fire location data were obtained from NASA Fire Information for Resource Management System (FIRMS) site (https://firms.modaps.eosdis.nasa.gov/firemap/) using the MODIS NRT C5 product.

### 2.3 In-situ observations

In situ data from the Mexico City area are used to provide context for the satellite measurements. We consider CO, as a tracer for combustion, and particulate matter (PM), both $PM_{2.5}$, which consists of particles with aerodynamic diameter less than 2.5 microns, and $PM_{10}$, defined as particles aerodynamic with aerodynamic diameter less than 10 microns. These data were obtained from 28 stations (Figure 2) of the Air Quality Monitoring Network of Mexico City (AQMN_MC) (Tzompa-Sosa et al., 2016) ((http://www.aire.cdmx.gob.mx/default.php?opc=%27aKBhnmI=%27&opcion=Zg==). Precipitation data were obtained from daily precipitation accumulation from 78 rain gauges across MCMA from the Pluviometric Network of Mexico City (http://www.sacmex.cdmx.gob.mx/sacmex/). We will also refer to an analysis carried out by Tzompa-Sosa et al. (2016), which used measurements of levoglucosan (LEV) and water soluble organic carbon (WSOC) from these same sites. The mass ratio of LEV to WSOC determined from source samples can be used to determine the contribution of primary biomass smoke to the total WSOC concentrations in ambient samples (Sullivan et al., 2008).

### 2.5 Model output

We used model output to provide background information and context for the TES measurements. HYSPLIT runs provided back trajectories, the MIROC model (see section 2.5.2) yielded a regional view of $O_3$ and $CH_3OH$ in the MCMA area, and the nested GEOS-Chem runs provided $O_3$, $NH_3$, $CH_3OH$ and $HCOOH$ at higher spatial resolution in the Lagos area.

### 2.5.1 HYSPLIT runs

To explore the sources influencing the trace gas mixing ratios measured by TES over Mexico City and Lagos we generated back trajectories using Version 4 of Hybrid Single-Particle Lagrangian Integrated Trajectory (HYSPLIT) model (Draxler and Hess, 1998). We used 1 deg x 1 deg gridded meteorological inputs from the Global Data Assimilation System (GDAS) to generate 5-day back trajectories from Mexico City and 10-day back trajectories from Lagos. The longer back trajectory length over Lagos was used to account for the slower wind speeds near the equator. The back trajectories at each location were initiated from 100, 500, 1000, 2000, 3000, and 4000 m above ground level to identify the potentially different source locations impacting the vertical profiles of pollutants in the lower troposphere.

### 2. 5.2 MIROC model output

The MIROC-Chem model (Watanabe et al., 2011) estimates detailed photochemistry in the troposphere and stratosphere by simulating tracer transport, wet and dry deposition, and emissions, and calculates

the concentrations of 92 chemical species and 262 chemical reactions. Its tropospheric chemistry was developed based on the CHASER model (Sudo et al., 2002) and it resolves the fundamental chemical cycle of Ox-NOx-HOx-CH$_4$-CO along with oxidation of NMVOCs to simulate ozone chemistry in the troposphere. However, NH$_3$ is not accurately modeled and HCOOH is not modeled separately, therefore

we will not cmpare MIROC output with TES data. Its stratospheric chemistry was developed based on the CCSR/NIES stratospheric chemistry model (Akiyoshi et al., 2004). MIROC-Chem has a T42 horizontal resolution (approximately 2.8°x2.8°) with 32 vertical levels from the surface to 4.4 hPa. It is coupled to the atmospheric general circulation model MIROC-AGCM version 4 (Watanabe et al., 2011). The MIROC-AGCM fields were nudged toward the 6-hourly ERA-Interim reanalysis (Dee et al.,

2011). The emission inventories used were obtained from the emission scenarios for Greenhouse Gas and Air Pollution Interactions and Synergies (GAINS) model developed by International Institute for Applied System Analysis (IIASA) (Klimont et al., 2009; Akimoto et al., 2015).

In this work the model employed daily surface NOx and CO emission data optimized using satellite measurements (OMI, GOME-2, and SCIAMACHY NO$_2$ measurements for NOx emissions, and

MOPITT CO measurements for CO emissions) for May 2013 from a data assimilation calculation (Miyazaki et al., 2016). The optimized emissions are about three times larger over Mexico city and about 20% larger for the United States total emissions in May 2013 than the a priori emissions constructed based on bottom-up emission inventrories from the Emission Database for Global Atmospheric Research (EDGAR) version 4.2 (EC JRC/PBL, 2012), the Global Fire Emissions Database

(GFED) version 3.1 (van der Werf et al., 2010), and the Global Emissions Inventory Activity (GEIA) (Yienger and Levy, 1995).  These results suggest that these global bottom-up emission inventories underestimate NOx emissions and may lead to a large uncertainty in ozone simulations around Mexico City.

### 2.5.3 GEOS-Chem output

The GEOS-Chem chemical transport model (version 10-01; http://acmg.seas.harvard.edu/geos/) uses as input GEOS-5 assimilated meteorology from the NASA Global Modeling and Assimilation Office (GMAO) at $0.5^0$x$0.667^0$ horizontal resolution nested over the African continent. Boundary conditions

are from a global simulation at $2^0$x$2.5^0$. The model includes detailed NO$_x$-VOC-ozone-aerosol photochemistry described in Mao et al. (2010) and updated here to include sea-air exchange of methanol as in Wells et al. (2014). Pollution sources relevant to Lagos, Nigeria include seasonal biomass burning from GFED4 (van der Werf et al., 2010), industrial emissions from EDGAR v4.2 (EC-JRC/PBL, 2011) for NO$_x$ and CO and RETROv2 for VOCs (Schultz et al, 2007), NH$_3$ from agricultural

activity from EDGAR v4.2, pollution from residential biofuel (fuelwood, charcoal, crop residue) use, charcoal production, backup generators, cars, and motorcycles from the regional DICE-Africa inventory

(Marais and Wiedinmyer, 2016), and trash burning from Wiedinmyer et al. (2014). Data are for year 2012 with one year of spinup for chemical initialization.

## 3 TES Megacity data examples

### 3.1 Mexico City

The Mexico City Metropolitan Area (MCMA) is home to over 20 million people and is situated in a basin 2240 m above sea level, surrounded by mountains on three sides. To the north lies the Mexican Plateau, while there is a small gap in the southeastern rim. At approximately 19.4°N, it is at the edge of the tropics and receives strong solar radiation that drives mountain-valley and urban-induced winds (de Foy et al., 2008), leading to convergent drainage flow and associated high concentrations of pollutants

(Jauregui, 1988). Data from aircraft campaigns have suggested Mexico City pollution may be linked biomass burning emissions originating outside the metropolitan area (Yokelson et al., 2011) and possibly even from the Yucatan peninsula (Kreidenweis et al., 2001). However, MOZART-4 model simulations indicate that emissions from open fires probably do not have a major impact on monthly average pollution within the city (Emmons et al., 2010). Here we use TES data to look for connections

between biomass burning and pollution in the MCMA (Figure 3) during the 2013-2015 period. The time series in the left panel of Figure 3 shows a distinct seasonal variability in all four species we are analyzing, with enhanced values between March and June of each year. This period coincides with the dry season and concomitant biomass burning events in Mexico, as shown by the time series of AIRS CO, a marker for combustion, and MODIS AOD; high aerosol AOD may indicate the presence of

smoke and/or dust (right panel of Figure 3). The seasonal cycle in the latter two datasets is qualitatively in good agreement with the cycle presented in all four TES time series.

The strongest peak in the time series for all species occurs on May 9, 2013 (indicated by red circles in Figure 3). This extreme event detected by TES occurs within a period of several days of maximum values in the AIRS and MODIS record for the 2013-2015 period. The concurrent peaks in CO, AOD,

and HCOOH, which is a known product of biomass burning, all strongly indicate biomass burning influence in the MCMA on this day, as does the very high of HCOOH (nearly 5 ppbv). We compared the wind direction and fire locations for May 9 with those on May 25, 2013 (green circle in Figure 3), the next TES MCMA transect, when abundances of all of the trace gases were much lower (Figure 4, top and middle panel). Back trajectories were calculated with the NOAA HYSPLIT model, while the

fire locations were obtained from the NASA FIRMS site.  The back trajectories for May 9 show that the air masses arriving on that day in MCMA passed over or near a number of fires west and southwest of MCMA. Conversely, the trajectories on May 25 mainly traversed the Gulf of Mexico over a region with few detectable fires before climbing to the MCMA plateau. The combined satellite data and back trajectories suggest that the air quality along the transect on May 9 was likely impacted by biomass

burning, but on May 25 these same data, in particular the ~3 ppbv lower value of HCOOH, indicate that transport from biomass burning emissions probably had much less of an impact, with local production possibly being responsible for the lower but still above average abundances detected on this day. The TES measurement on April 23 (blue circle in Figure 3) is interesting in that it contains elevated $O_3$,

$CH_3OH$ and HCOOH, all of which could have a fire source, but low $NH_3$. The fire map and back trajectory for this day (Figure 4, bottom panel) show winds from the same direction as on May 9, but somewhat fewer fires;, the low values of $NH_3$, which has a shorter life than the other species, could indicate that these fires were further away or weaker than on May 9. By June 10 (brown circle in Figure 3) the TES transect (the second point after the May 9 peak), shows low values for all species, indicating

cleaner air.

Surface measurements of precipitation, $PM_{2.5}$, $PM_{10}$ and CO (Figure 5) show that the late April-early June period in 2013 can be divided into two regimes:  a dry regime from April 27 through May 11, with higher concentrations of $PM_{2.5}$, $PM_{10}$ and CO, peaking around May 9, followed by a wet regime through June 2, with lower pollutant concentrations and a number of rainy days. Tzompa-Sosa et al.

(2016) calculated LEV/WSOC ratios during May 2013, which provide an estimate of the contribution of biomass burning to the total WSOC in an air mass. Their results show higher LEV/WSOC ratios during the first weeks of May, compared to the rest of the month. Taken altogether, the data indicate that biomass burning contributes much more to pollution on May 9 than May 25, due to a combination of transport and the lack of precipitation to drive wet deposition.  It is, in fact, likely that the changing

meteorological conditions associated with the wetter weather were also responsible for the shift in transport away from the fire locations. Note that on June 10[th], when TES data suggest cleaner air, the in situ measurements show rain and lower values of CO and particulate matter.

However, it is important to note the impact of biomass burning varies spatially. Figure 6 shows that $NH_3$ and HCOOH concentrations are extremely high to the south of the Mexico City basin on May 9

and drop off markedly as TES moves north of 19.1° and elevation increases; on May 25 the decrease in $NH_3$ and HCOOH with elevation is much more gradual, and there is actually an increase as the TES observes the air mass inside the basin. The overall abundances of $NH_3$ and HCOOH are higher on May 9 than May 25, which is consistent with the in situ observations, but the difference between the days is much more marked south of the caldera. These results suggest that on May 9 the air inside the basin was

somewhat isolated from the biomass burning influence due to the fact that the air from the fires had to travel from the southwest up and over the mountains to reach MCMA (Figure 4, top panel). Seasonal means of $NH_3$ and HCOOH (Figure 7) along the transect path show that this a regular feature, especially in the December through May period:  concentrations decrease sharply as the transect passes north over the highest elevation point, at 3088 m, just south of Mexico City, though the MAM HCOOH

mean reaches its minimum slightly further north. This result is in agreement with the Emmons et al.

(2010) model output, which shows only a a weak influence of biomass burning on the average pollution within the city.

Both $O_3$ and $CH_3OH$ are generally higher on May 9 than on May 25, confirming the high pollution level of this day as indicated by $NH_3$ and HCOOH. On the other hand, the spatial variability along the transects for $O_3$ and $CH_3OH$ is very different from that of $NH_3$ and HCOOH. On May 9 $O_3$ is relatively flat, while $CH_3OH$ has higher values and variability within the caldera rather than to the south. This could point to enhanced local production within the city of both gases on this very polluted day, combined with a large scale event that involves transport of $O_3$.

We examined the capability of the MIROC model to capture the different conditions on these two days. Due to its coarse resolution MIROC cannot model the spatial variability seen in the TES data, so we compared the average over the transects against the model grid cell containing Mexico City. On May 9 (Figure 8, top panels) TES data show that above 800 mbar the model severely underestimates both species on this day. These underestimations could be attributed to transport errors associated with the relatively coarse model resolution (i.e., 2.8°) and to errors in photochemical productions for such an extremely polluted case. On May 25 (Figure 8, bottom panels), when biomass burning appears to play a much smaller role, there is better agreement between TES and the MIROC profiles, though either the model has overestimated $CH_3OH$ below 800 mbar or TES was not sufficiently sensitive to $CH_3OH$ at this altitude range. $O_3$ is slightly overestimated below 600 mbar. The overall good model performance is largely attributable to the use of optimized anthropogenic and biomass burning emissions of NOx and CO using satellite measurements (Miyazaki et al., 2016), but this example demonstrates the value of the TES data in capturing events at finer scales than large scales are capable of estimating.

## 3.2 Lagos

The atmosphere above urban West Africa is very poorly understood, in spite of the large and growing population. Not only do the pollution levels impact the health of tens of millions of people, they very likely alter the precipitation patterns of the West African Monsoon (Knippertz et al. 2015). While there have been a number of measurement campaigns in the region, such as the African Monsoon Multidisciplinary Analysis (AMMA) (Mari et al., 2011), the IGAC (International Global Atmospheric Chemistry)/DEBITS (Deposition of Biogeochemically Important Trace Species)/AFRICA atmospheric chemistry and deposition monitoring network IDAF (http://idaf.sedoo.fr; in operation since 1995, Liousse et al., 2010) and the Aerosol Robotic Network (AERONET), these have focused on natural emissions in remote locations. For example, the closest AERONET station to Lagos is at Ilorin, almost 300 km away in the sub-Sahel. There is very little in situ monitoring within the Lagos region. This unavailability of local data increases the value of the TES data, as it provides consistent (i.e., from the same platform at the same time) and repeated (every two weeks) measurements of a number of species

over a 200 km extent. The TES data can can provide a broader context for the limited in situ measurements available, and point to interesting processes and trends, which can be compared to those from the in situ data.

Lagos is a tropical city at 6.5° N on the northern edge of the Gulf of Guinea. With over 12 million people, Lagos is Nigeria's and Africa's largest city and one of the fastest growing megacities, with a growth rate currently at 4%/year (UN, 2014). This rapid growth has had many consequences, including traffic congestion, high use of very inefficient motorcycles with two-stroke engines, unreliable electrical supply, and poor waste disposal, which in turn have led to high emissions from vehicles, diesel powered generators and waste incineration (Hopkins et al., 2009; Marais et al., 2014). The oil and gas extraction activities in the Niger Delta to the southeast also contribute to poor air quality through flaring, gas leakage and pipeline explosions (Marais et al., 2014). At this low latitude synoptic variability is limited. Air quality in the Lagos region is strongly affected by the West African Monsoon (WAM) during the wet season (June-August (JJA)), which effectively ventilates the area (Marais et al., 2014) and reduces the aerosol load through precipitation (Knippertz et al., 2015). In the dry season (December-February), strong temperature inversions at 900-750 hPa inland, caused by the warm northeasterly Harmattan winds and the blocking anticyclone from the Sahara (Sauvage et al., 2005), trap pollutants in the lowest levels of the atmosphere. The ventilation levels during this season are among the weakest in the world (Marais, 2014). There is frequent recirculation of air due to sea breezes, an example of which is shown in the back trajectory maps for February 7, 2013 (Figure 9). Anthropogenic pollutants can build up and interact with the products from frequent biomass burning events (Figure 9). Sunlight driven chemistry can produce ozone and acids ($H_2SO_4$, $HNO_3$); the latter react with ammonia to create $PM_{2.5}$ particles which contribute to the aerosol loading.

TES transects were carried out slightly to the west of the Lagos area (Figure 10); the southern end of the transect, south of 6.2°N, was over the Gulf of Guinea. The elevation is low and constant; there is no complicated topography affecting winds and transport as there is in the MCMA, thus we expect that emissions from the city will strongly impact the surrounding areas, and that the TES measurements will provide information on pollution levels in the greater Lagos metropolitan area. Means of the TES observations along the entire extent of the transect (Figure 11, left panel, solid line) show a strong seasonal pattern in all four species, with clear enhancements during the December-March period, which correlate with the AIRS CO and MODIS AOD seasonal variability (Figure 11, right panel). Furthermore, in 2015 MODIS AOD was higher than in 2013 and 2014, and these elevated amounts persisted until June. Similar persistence is seen in the TES $O_3$, which captured three extreme events with $O_3$ greater than 100 ppbv, and in the TES HCOOH data. Comparing the Lagos time series to those from the the MCMA (Figure 11, left panel, dashed line), it becomes obvious that the air quality problems of these two regions are different in magnitude. Mexico City has a reputation for severe pollution, yet the high levels of all four trace gases measured during the extreme event on May 9 in the

MCMA are actually at or below the all the measurements during the dry season levels in the Lagos area, likely reflecting the greater number and strength of pollution sources, from biomass burning, traffic, power generation and the petrochemical industry, combined with the stagnation and recirculation described above. This demonstrates the potential of the TES megacity data to quickly evaluate the similarities and differences in the air quality components of the world's megacities.

Further evidence of the recirculation of trace gases is provided in the spatial variability of the seasonal means over the 2013-2015 period (Figure 12, top row). During the DJF period the values over the Gulf are approximately 90 ppbv, 4 ppbv, 4 ppbv and 4.5 ppbv for O3, NH$_3$, CH$_3$OH and HCOOH, respectively; these values are a great deal higher than typical remote ocean values (i.e., 30 ppbv, 0.1 ppbv, 1.0 ppbv, 0.01 ppbv from GEOS-Chem global runs) for the same species, and are in general as high as (except for NH$_3$) the amounts over land, even though there are no significant sources of any of these species in the Gulf. NH$_3$ does show a sharp decrease as the transect crosses land to ocean: this may be due to some of the NH$_3$ carried out to sea reacting with the nitric and sulphuric acids derived from NO$_x$ and SO$_2$ emissions, or being deposited to the ocean surface. NH$_3$ also exhibits this spatial gradient in the other three seasons, especially the MAM period. In every season NH$_3$ peaks north of the Lagos area, possibly due to contributions from both agricultural activities and biomass burning. On the other hand, CH$_3$OH and HCOOH show steep decreases in JJA and SON as the transects moves north of 7°N, into less urbanized areas: the main source for these gases during the rainy season, in absence of biomass burning, are likely emissions in Lagos proper, and wet deposition may removes them before they reach this region. There is also a dip in the O$_3$ concentrations in DJF and MAM between 6.4°N and 7°N, as the transect approaches the coast and passes along the Lagos urban area, possibly due to higher NO$_x$ trapped in the boundary layer. The seasonality in the TES measurements is also evident in these figures; the highest values are observed in DJF and MAM, with little difference between the two periods for CH$_3$OH and HCOOH, and larger differences for O$_3$ and NH$_3$. Measurements are much lower in JJA and SON seasons.

Comparisons with model output are necessarily qualitative, as the fire emissions and meteorological conditions between the GEOS-Chem 2012 runs and the TES 2013-2015 measurements are likely quantitatively different. Uncertainties in the fire emisisons used in the model are likely greater than the interannual variability, and the TES spatial variability should be higher, as the TES pixel is twenty five times smaller than the GEOS-Chem nested grid box. However, the seasonal patterns would be expected to show similarities; both AIRS CO and MODIS AOD from 2012 (Figure A3) present peak values in January-February, as is seen in the 2013-2015 period; the MODIS AOD in 2012 is at the high level observed in 2015, while the AIRS CO in 2012 is similar to the CO levels observed in 2013-2015. The GEOS-Chem spatial plots (Figure 12, bottom row) were obtained by averaging the GEOS-Chem mean seasonal profile values over the same vertical range used to provide the TES spatial plots. This vertical

range varied slightly by season and latitude, due to variability in the TES sensitivity for all species except $O_3$, for which the range was fixed to cover the lowest three measurement levels. We used the TES vertical averaging interval in order to reduce differences arising from averages over different numbers of levels. TES and GEOS-Chem both see high values in DJF, lower values in JJA and SON, and a latitudinal gradient in $NH_3$ in DJF, although the GEOS-Chem gradient is much weaker. GEOS-Chem predicts ocean-land gradients for all species and seasons, but these are only only observed in the TES $NH_3$ measurements, suggesting that the model may be underestimating transport of pollution off-shore, particularly during DJF and MAM, either because of differences in meterology between the modelled and measured time periods or because of deficiencies in modelling of the local scale land-sea circulation. GEOS-Chem values are lower by a factor of 2 (for $O_3$) to 10 (for $NH_3$), except for $CH_3OH$ over land in DJF, where both model and measurement predict ~ 4 to 5 ppbv. Chaliyakunnel et al. (2016) have shown that there are secondary sources of HCOOH from fires that are not well modelled, which could be contributing to the underestimate of HCOOH. However, since these large differences are evident in the HCCOH data in the non-burning season, and are also seen in $O_3$, and $NH_3$, the city of Lagos itself is probably a strong pollution source, due to the conditions described at the beginning of this section.

**4 Summary**

The TES Megacity dataset covers the period from January 2013 to present and provides air quality data over nineteen metropolitan areas. We focused on $O_3$, $NH_3$, $CH_3OH$, and HCOOH for their role in determining air quality.

In the Mexico City region the TES-observed seasonal signal is qualitatively well correlated with AIRS CO and MODIS AOD, with peaks in the drier MAM period, which is also the peak of the biomass burning season. A very high pollution event on May 9, 2013, was observed and attributed to contributions from biomass burning emissions transported from the southwest. This event was observed in Mexico City itself, where high LEV/WSOC ratios were measured, but the TES data showed it was significantly stronger south of the city. This pattern of higher levels of these trace gases south of the caldera was repeated in MAM over the 2013-2015 period monitored by TES. A weaker event on May 25, 2013, in a rainier period with easterly winds, showed lower biomass burning influence both in the TES data, which exhibited lower HCOOH levels than on May 25, and in the in situ LEV/WSOC ratios, which were also significantly lower.

The TES data over Lagos showed a strong seasonal signal correlated with AIRS CO and MODIS AOD, peaking in the dry DJF period, when biomass burning events are most frequent and western Africa often experiences stagnant air conditions. Except for $NH_3$, TES did not observe gradients between the Gulf and the coast, even though these were predicted by the GEOS-Chem model. As sources of all these species are weak over the ocean, the abundances observed by TES are likely due to stagnant air masses

being driven by sea breezes back and forth between the Gulf and the continent, suggesting that GEOS-Chem is not  modeling the local transport correctly. The levels of trace gases measured by TES were noticeably higher over Lagos than over Mexico City, and these higher levels were more persistent. This points to a difference in air quality between the two cities: while Mexico City has pollution events in MAM, Lagos has almost continuous high levels of pollution in DJFM.

In summary, the TES Megacity dataset can provide seasonal variability and spatial gradients of many species relevant to air quality for cities around the world. The data can be used for process studies and model evaluation, and allows for easy comparisons between two or more cities.

## Appendix

Infrared sensors on satellites have not traditionally been used for boundary layer measurements of $O_3$, given that they are most sensitive to $O_3$ in the mid-troposphere.  However, while they are less sensitive to surface amounts, they are are not insensitive. TES captures sufficient information from the boundary layer to clearly move the retrieval in the lowest three layers away from the a priori (Fig A1) during pollution events over the MCMA and the Lagos region, thus providing a signal that is linked to changes in surface concentrations.

We have shown that the levels of the trace gases measured by TES frequently are much higher over the Lagos region than over the MCMA (Section 3.2, Figure 11). This could be attributed to greater sensity of the TES retrievals over Lagos, but a comparison of the DOFS, which provide a measure of the information provided by the instrument (Fig A2), shows that the sensivity over MCMA overall and over Lagos during the dry season (roughly December to April) are comparable, with the exception of HCOOH, which has extremely enhanced values over Lagos during this period, and consequently a strong signal and elevated DOFS.

## Acknowledgements

We would like to thank all the people who collect and make public the in situ measurements from the Mexico City Metropolitan area, including rain gauge data. The air quality website maintained by Mexico City (http://www.aire.df.gob.mx/) is a model of ease of use and provides a wealth of information. Rick Pernak provided invaluable assistance with the revised figures. We would also like to thank Dylan Millet and Xi Chen for providing methanol and formic acid updates to GEOS-Chem. Funding for Zitely A. Tzompa-Sosa was provided by Consejo Nacional de Ciencia y Tecnología

(CONACYT) under fellowship No. 311461 and Mario Molina para Ciencias Ambientales fund. Part of this research was carried out at the Jet Propulsion Laboratory, California Institute of Technology, under a contract with the National Aeronautics and Space Administration. Copyright 2017.

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

# Figures

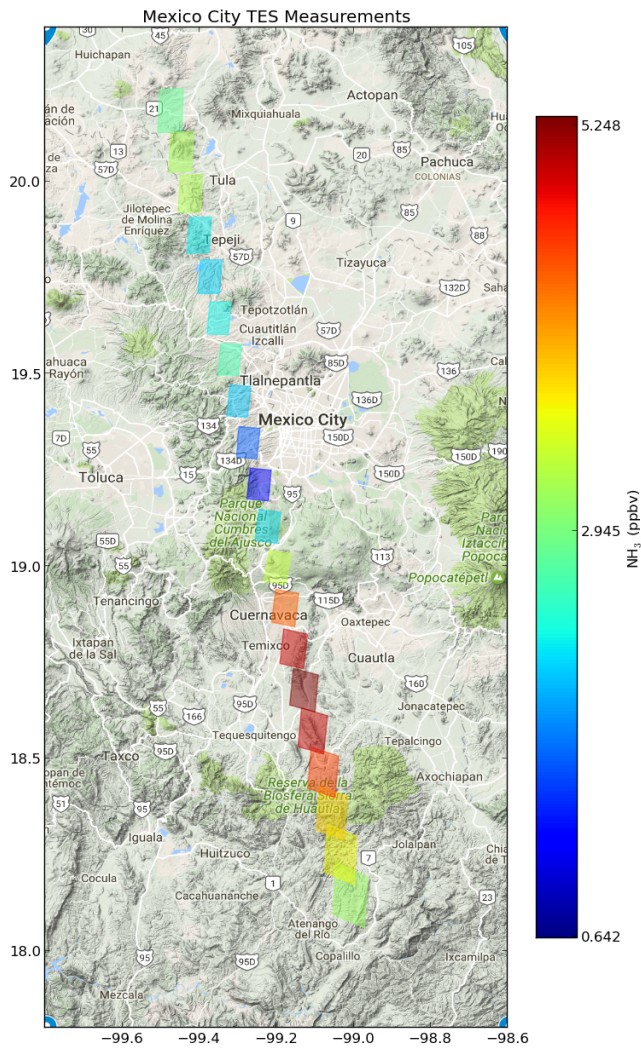

**Figure 1: Mean MAM TES NH$_3$ transect over the MCMA region.**

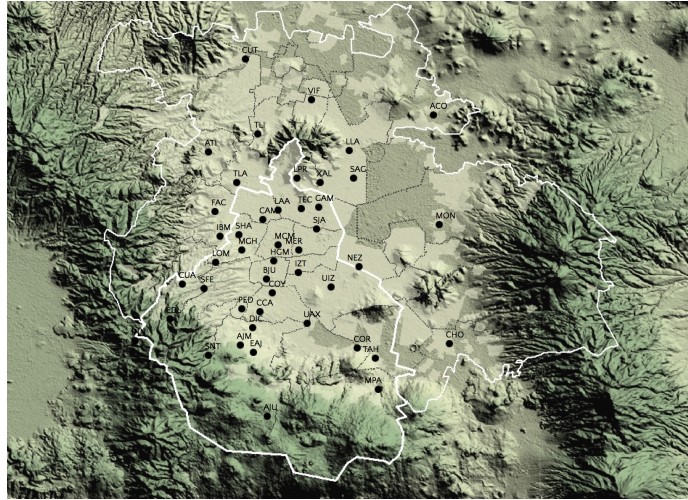

**Figure 2: Locations of monitoring sites in the MCMA area.**

**Figure 3:** (left) Means of the lower levels (see text for definition) of each TES transect over the MCMA area; circles mark dates discussed in text: blue: April 23, red: May 9, green: May 25, gold, June 10; (right) AIRS CO @850 hPa (top) and MODIS Deep Blue AOD (bottom) 1 deg x 1 deg data averaged over the grid boxes traversed by the transect.

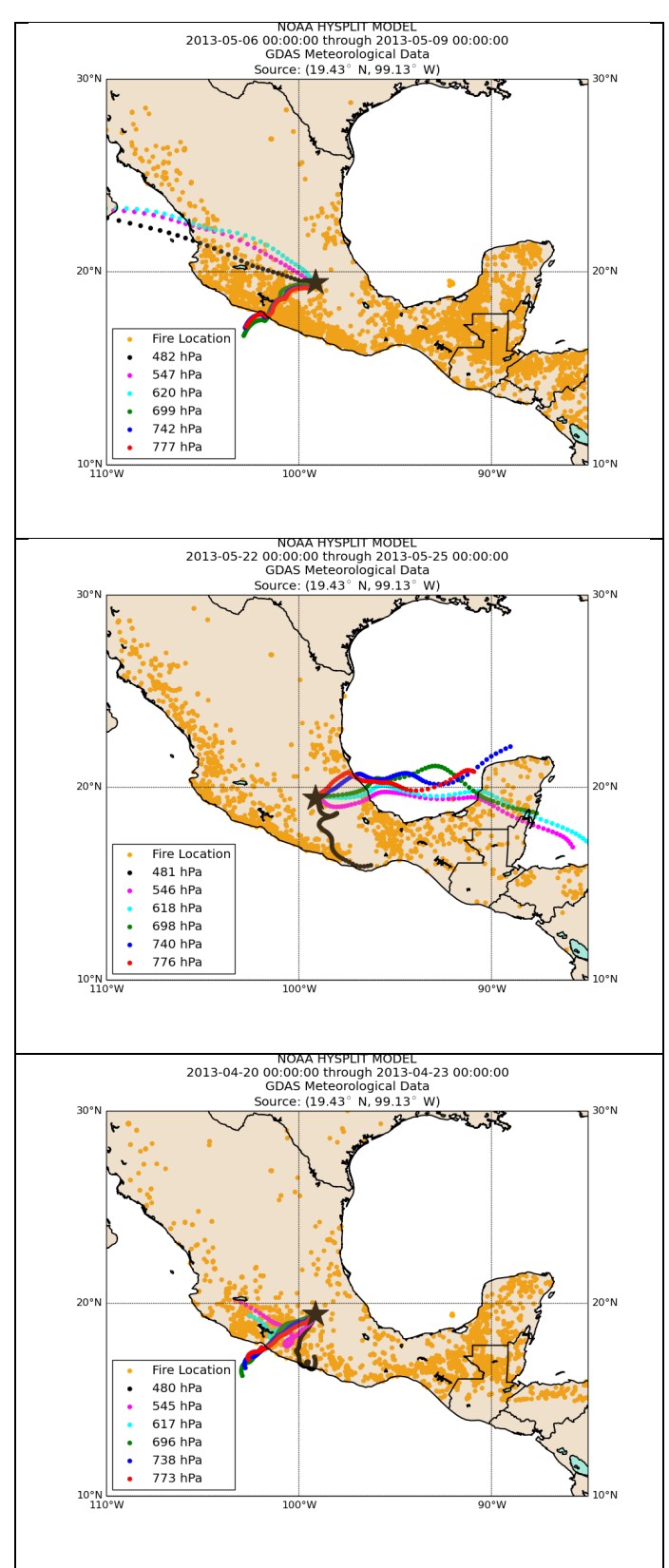

**Figure 4: HYSPLIT back trajectories and FIRMS fire locations in Mexico for May 9 (top), May 25 (middle), and April 23 (bottom). Fire maps include fires from the selected dates and the two previous days. Pressures on back trajectory plots refer to level at which the back trajectory was initiated.**

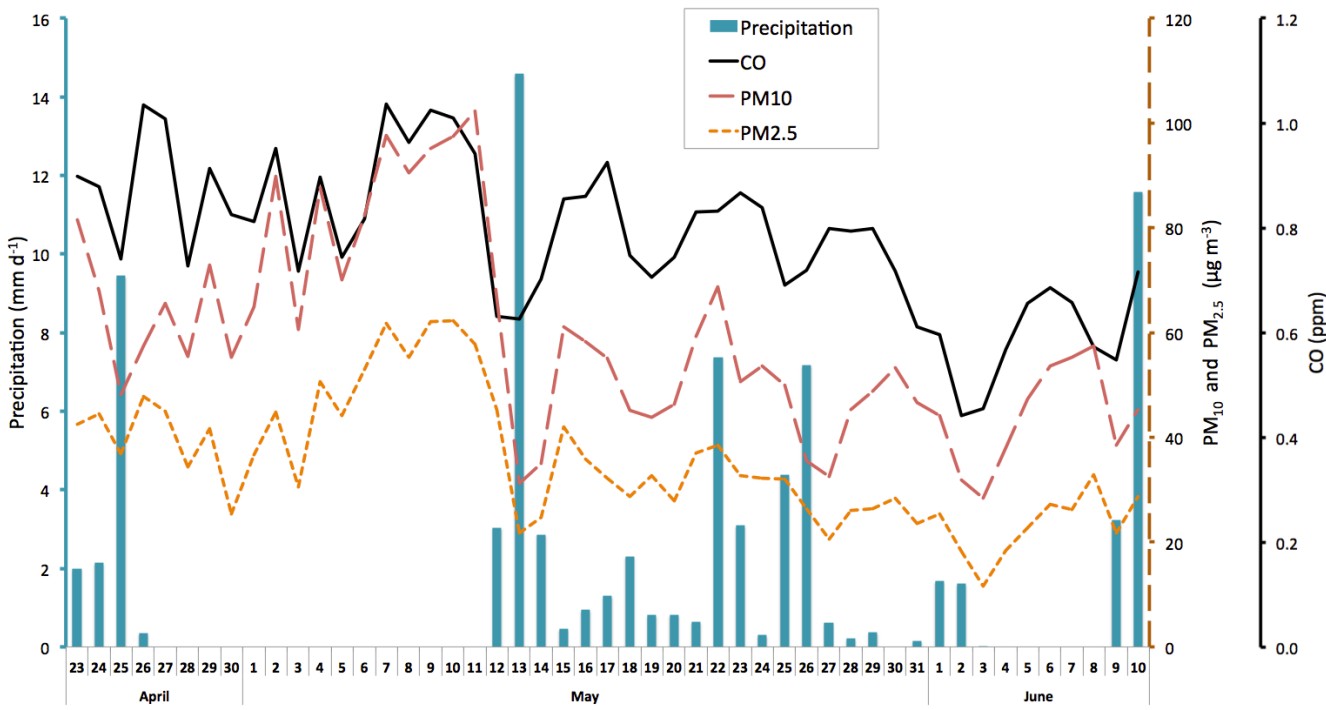

5 **Figure 5: Averaged daily precipitation and ambient concentrations of $PM_{2.5}$, $PM_{10}$, and CO over the MCMA. Averaged daily precipitation from 78 stations located inside the MCMA. Averaged ambient concentrations of $PM_{2.5}$, $PM_{10}$, and CO over the MCMA correspond to 28 stations located inside the MCMA (see Figure 2).**

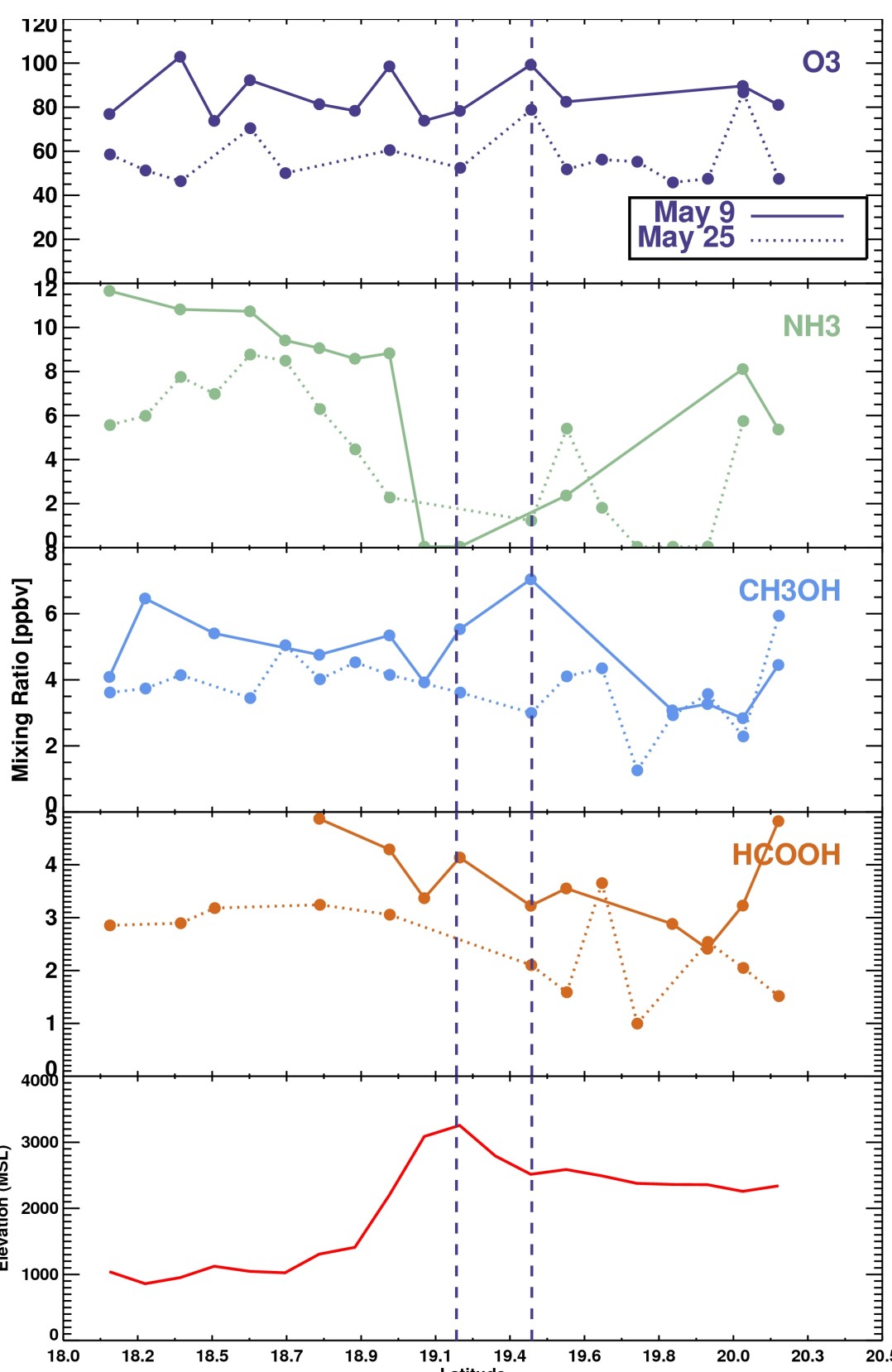

**Figure 6: TES retrievals versus latitude on May 9 (solid) and May 25 (dotted) 2013.: O₃ (top panel), NH₃(second panel), CH₃OH (third panel), HCOOH (fourth panel); elevation (bottom panel). Southern dashed line indicates latitude of the edge of the caldera containing Mexico City, where elevation is maximum; northern dashed line indicates center of Mexico City.**

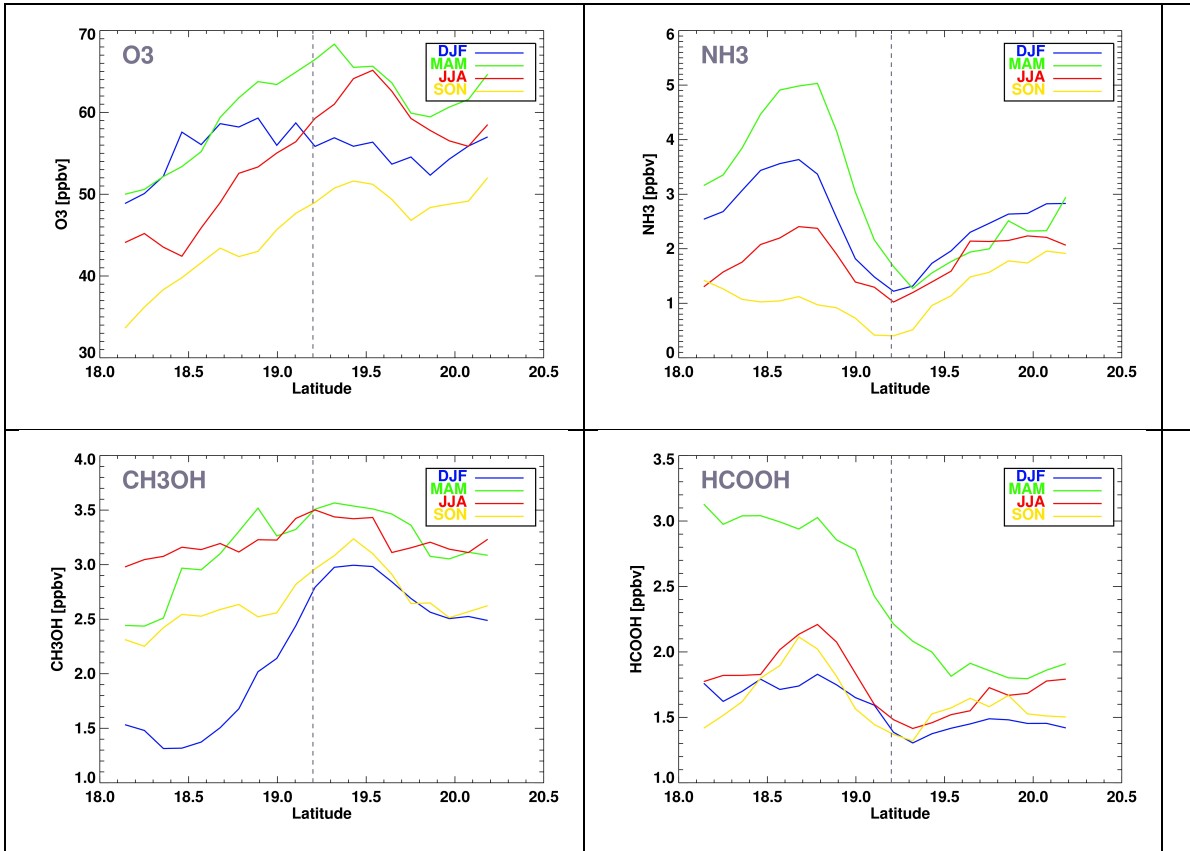

**Figure 7: Seasonal means along the MCMA transect: O₃ (top left), NH₃ (top right), CH₃OH (lower left), HCOOH (lower right).**

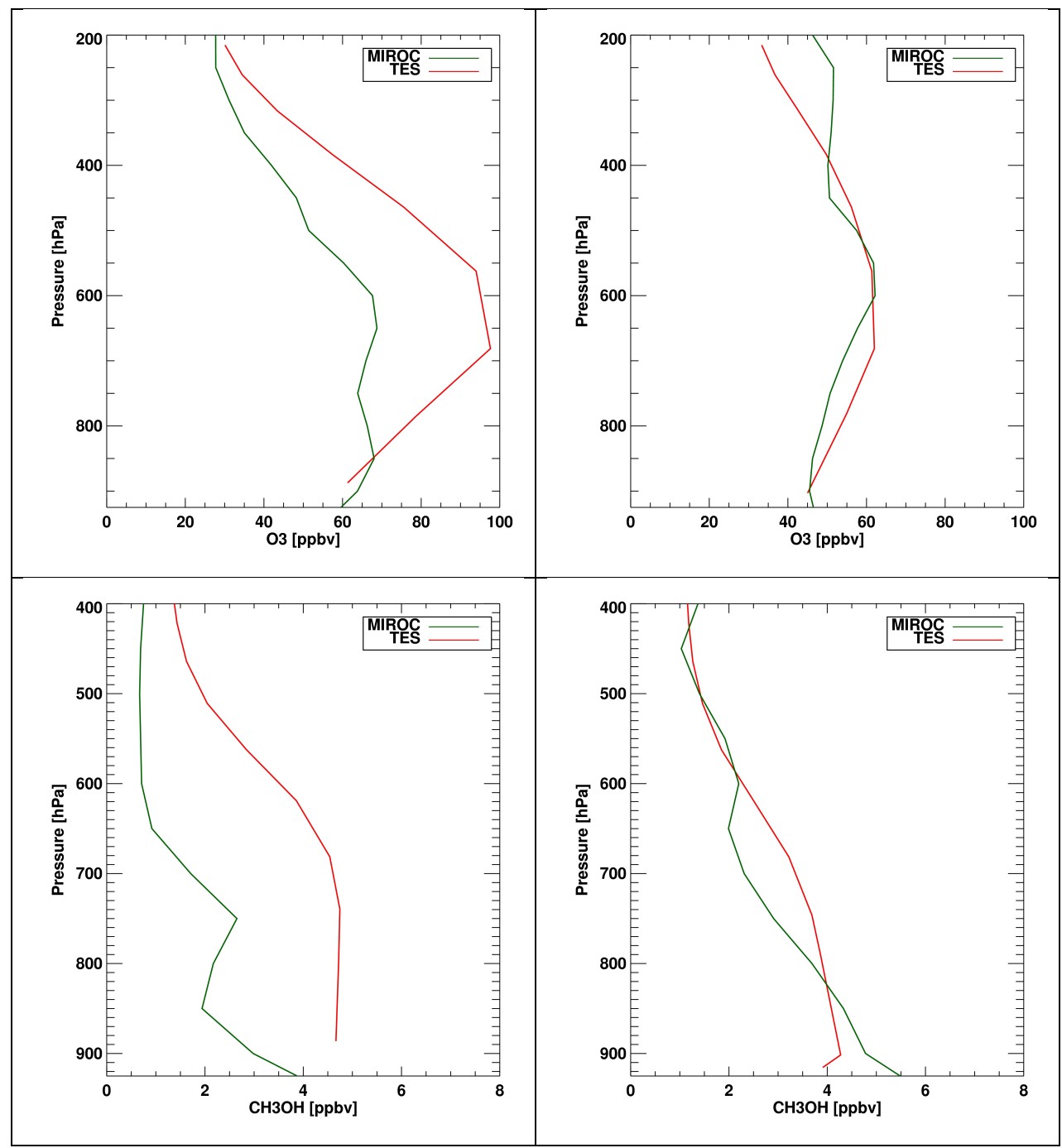

**Figure 8: MIROC model and TES retrieved profiles for O$_3$ (top) and CH$_3$OH (bottom) on May 9 (left) and May 25 (right).**

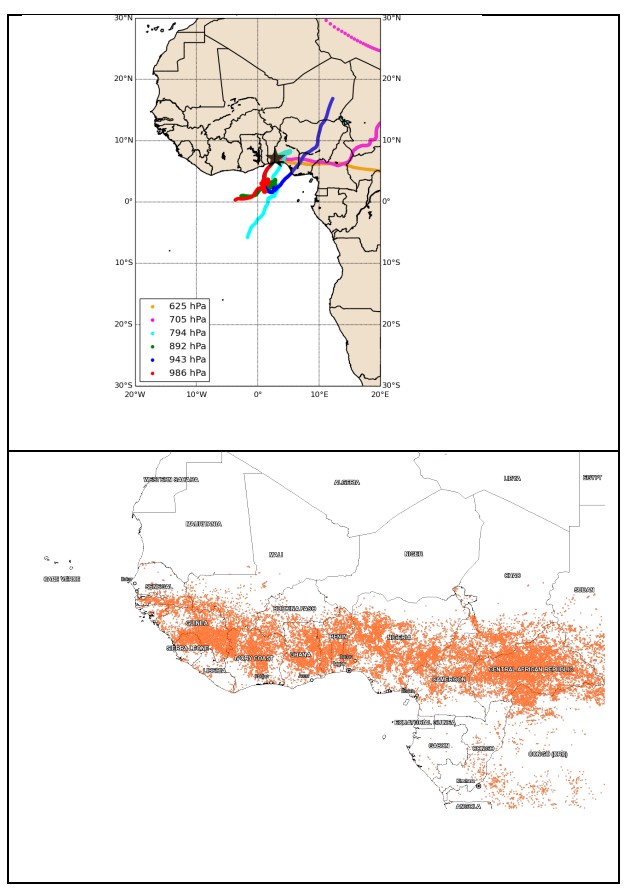

**Figure 9:    HYSPLIT back trajectories (top) and FIRMS fire locations (bottom) over western Africa for February 7, 2013. Fire map include fires from February 7 and the previous ten days.**

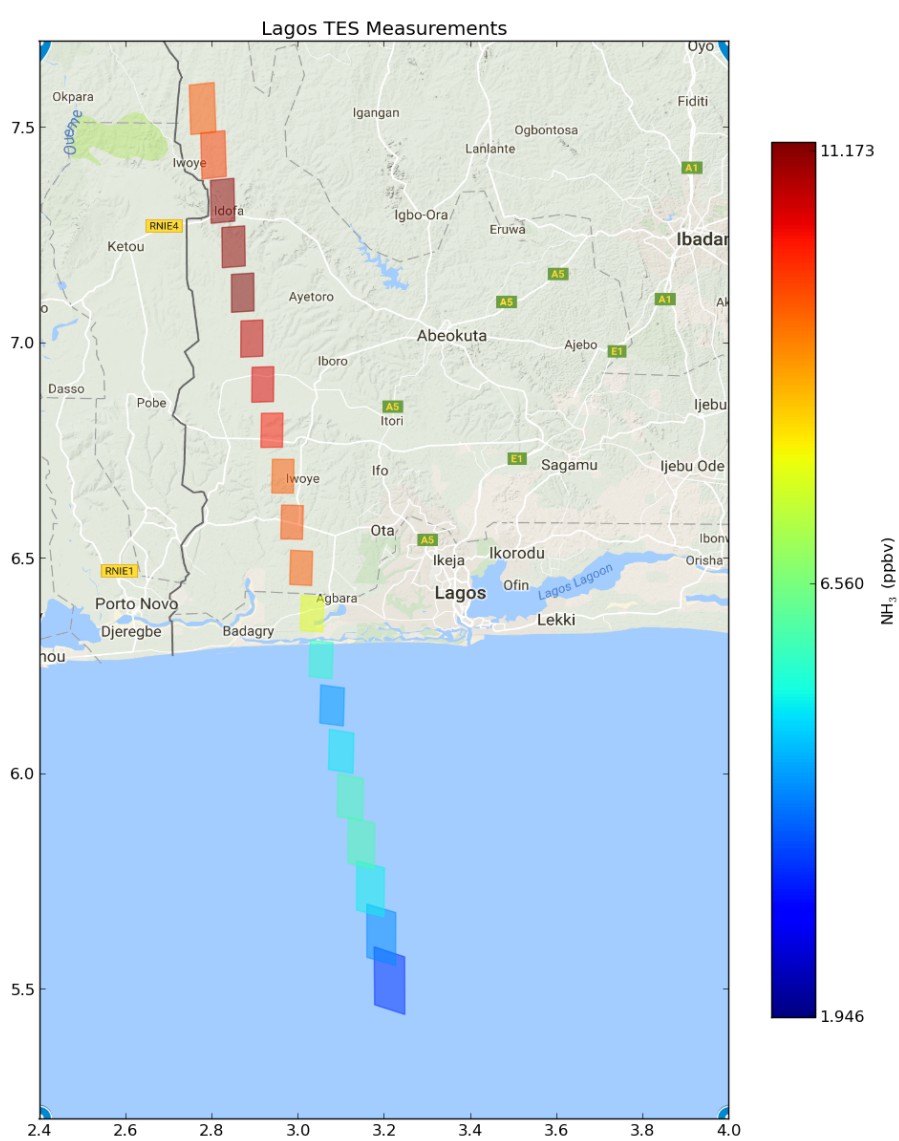

**Figure 10: Mean DJF  TES NH₃ transect over the Lagos region.**

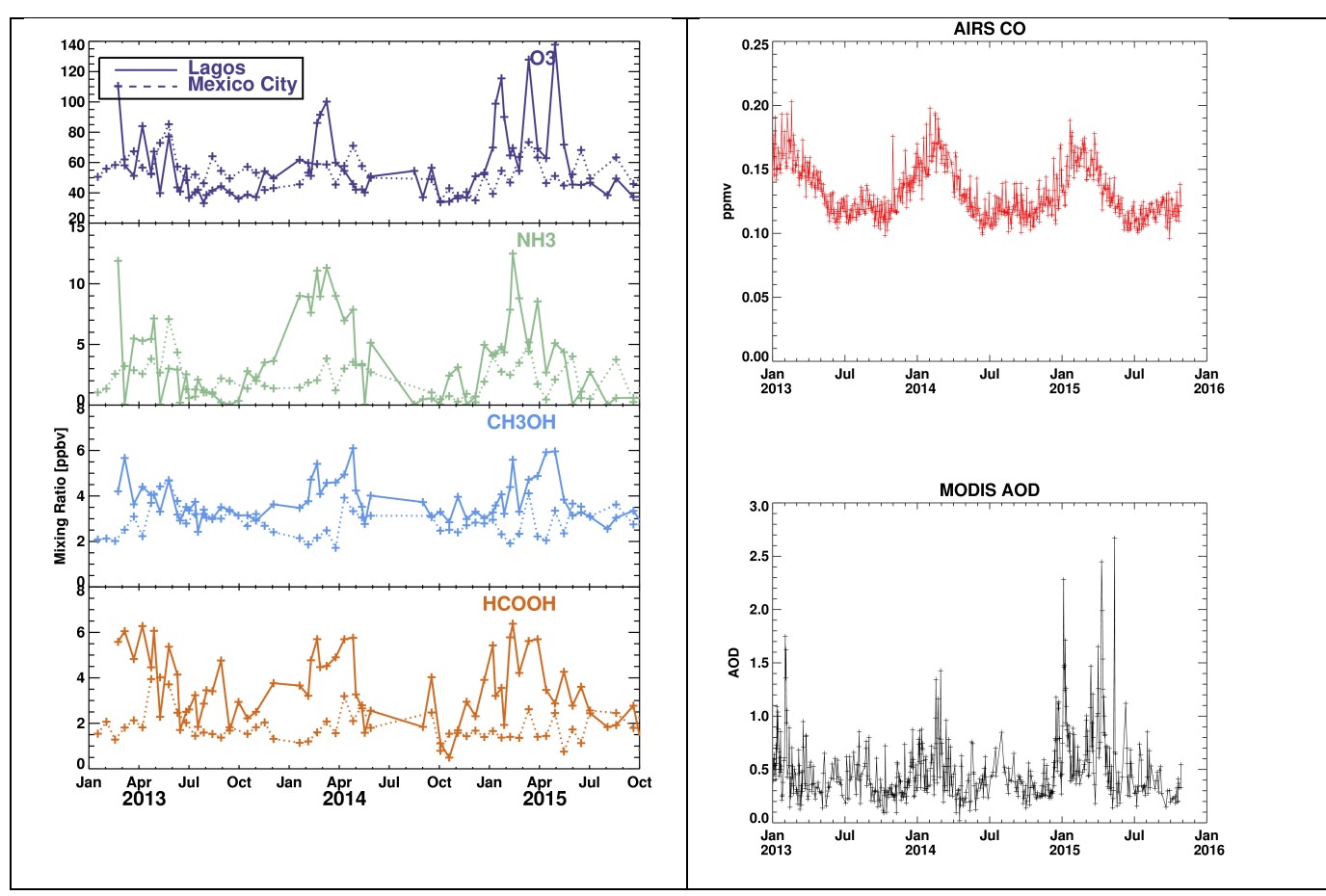

5 **Figure 11: Means of the lower levels of each transect over the Lagos transects area (solid) and MCMA (dashed); (right) AIRS CO @1000 hPa (top) and MODIS Deep Blue AOD (bottom) 1 deg x 1 deg data averaged over the grid boxes crossed by the Lagos transect.**

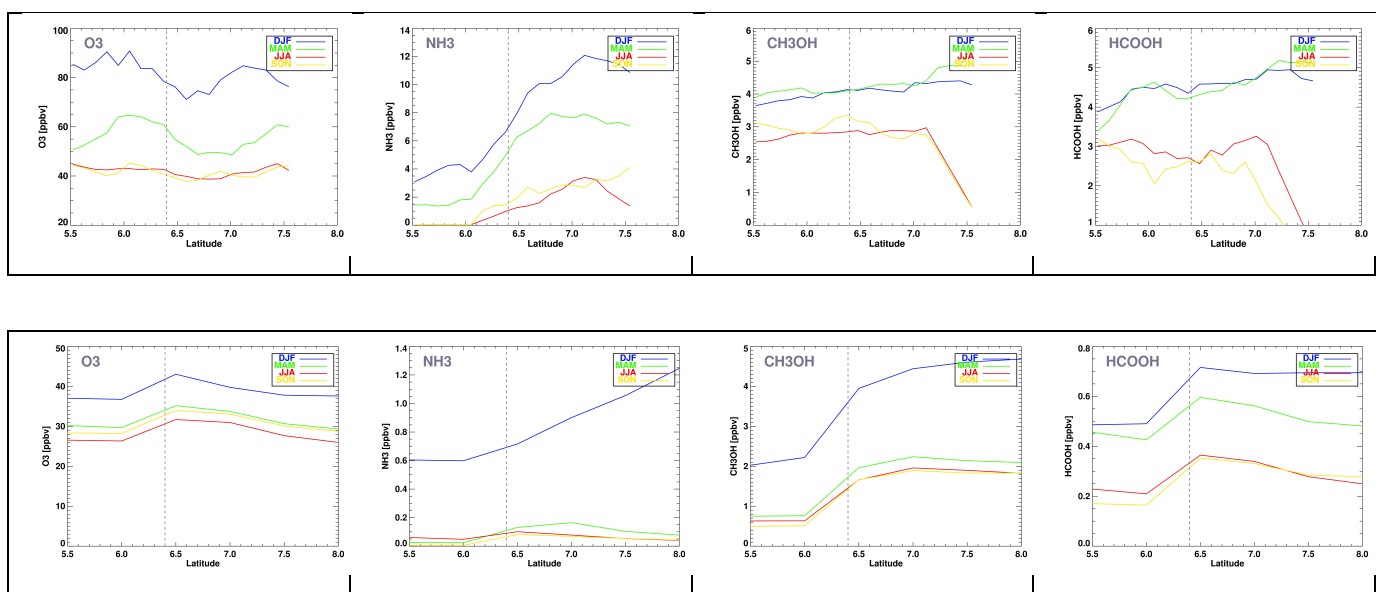

**Figure 12: Seasonal means along the Lagos transect. TES 2013-2015 means (top), GEOS-Chem 2012 means (bottom). Note different y-axis vertical range between TES and GEOS-Chem plots. Dashed line indicates latitude of land-sea interface.**

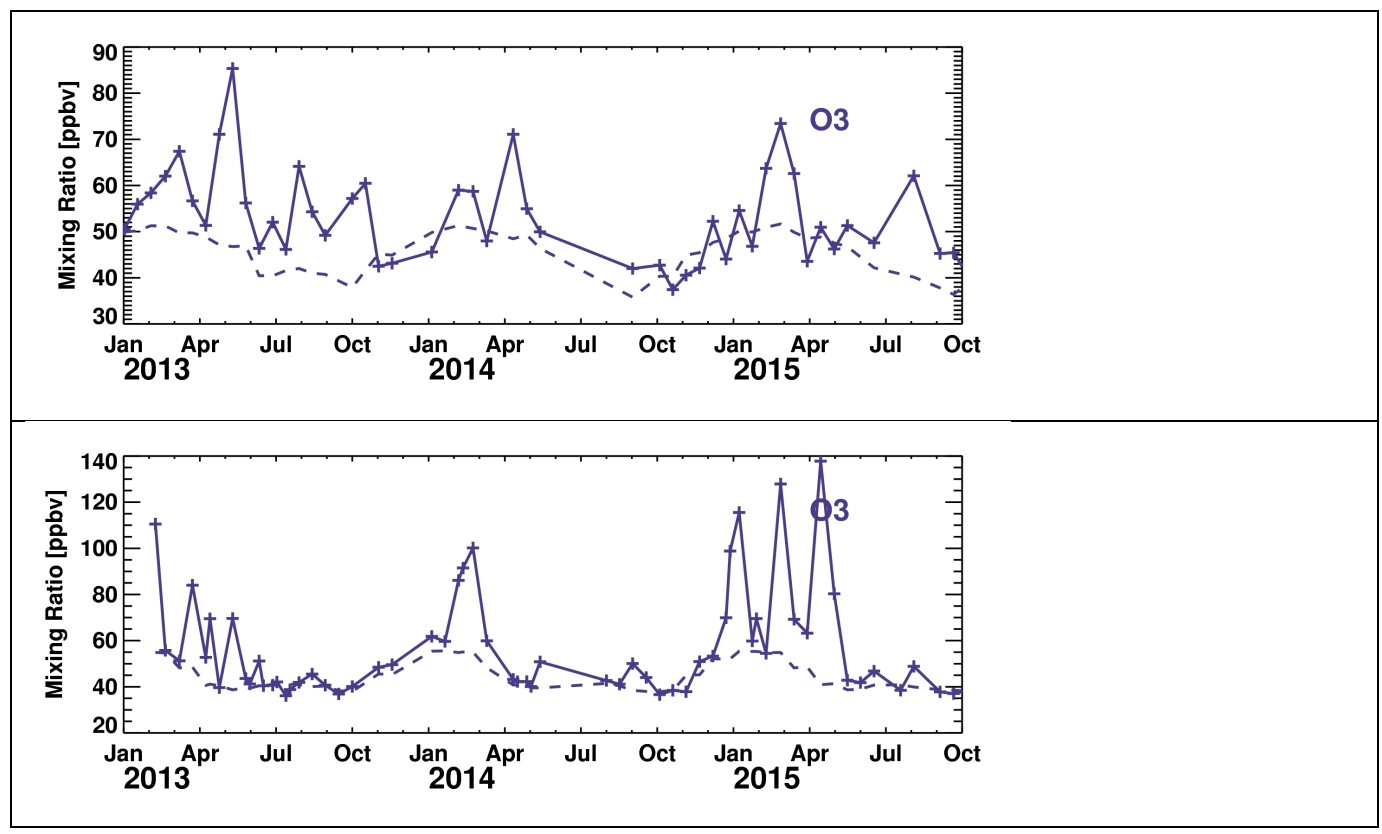

Figure A1: Time series of transect means (solid lines) and corresponding a priori means (dashed lines) for the MCMA (top) and the Lagos area (bottom).

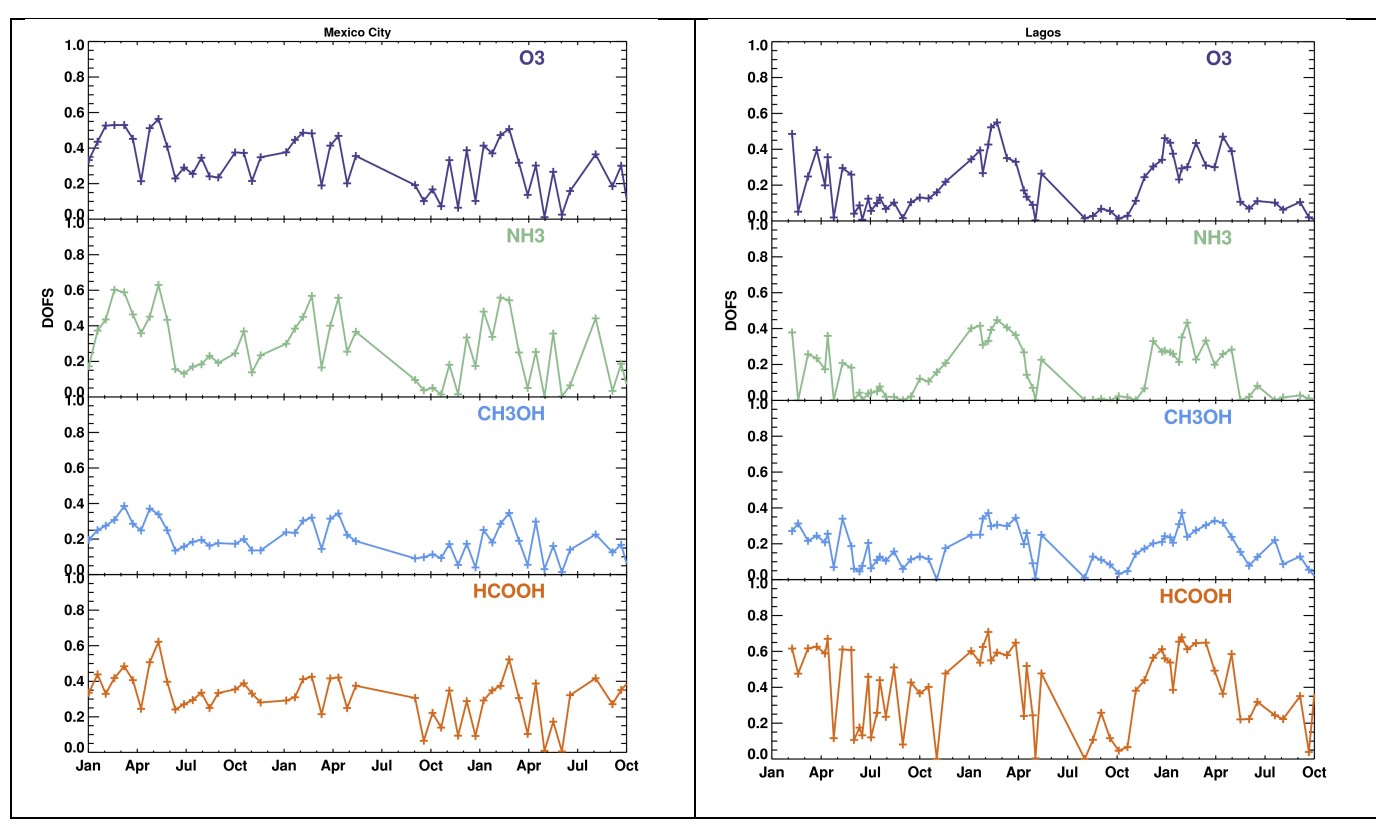

Figure A2: Time series of the DOFS over the MCMA (left) and the Lagos area (right).

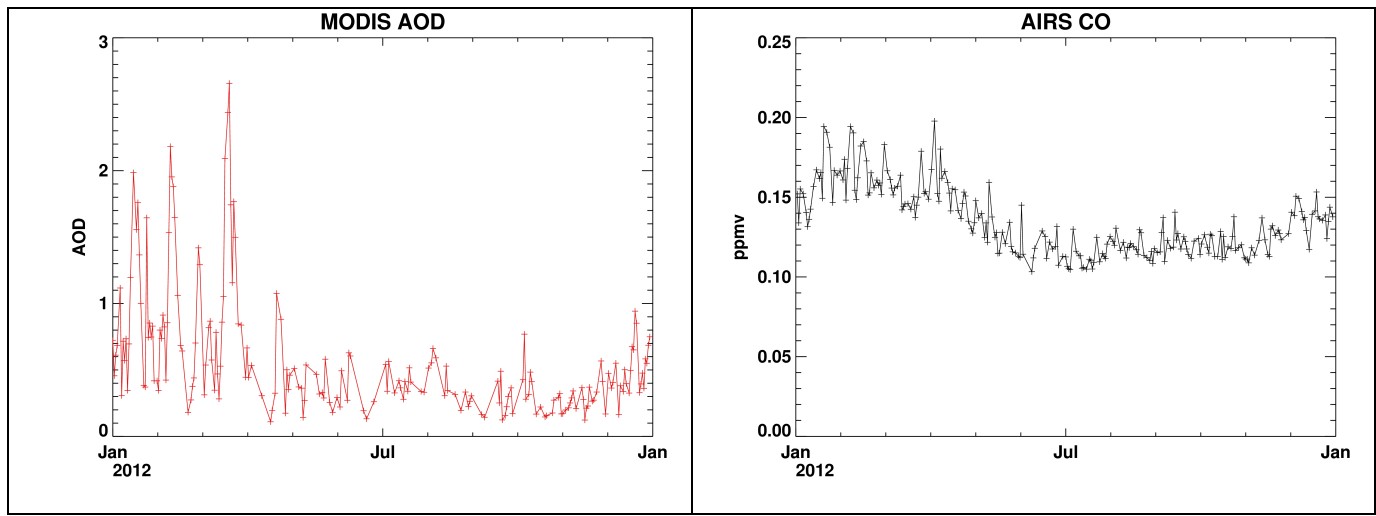

**Figure A3: 2012 time series over the Lagos area: MODIS AOD (left) and AIRS CO (right).**