# Peer review of "Seasonal and Spatial Changes in Trace Gases over Megacities from AURA TES Observations"

_Atmospheric Chemistry and Physics, 2017_

## Referee Comment (RC1) · Anonymous Referee #1 · 5 Apr 2017

This manuscript provided an interesting study about long-term observations of trace gases from TES over two megacities. TES observations, as well as satellite products and global model results, are capable to capture the seasonal signal, identify a pollution episode, and confirm the impacts from biomass burning on local air quality. The topic is applicable for Atmospheric Chemistry and Physics. The text is concisely written and well documented. Generally, the study about the Mexico City is very comprehensive including satellite observations, surface measurements, and model simulations. However, the case about Lagos lacks validation of satellite observations and model results due missing of in-situ measurements. The Mexico City and Lagos are substantially different in the chemistry and environment. For instance, Lagos has lots of emissions

from oil and natural gas industry, so it could be of problem to use the similar approach to analyze results from these two cities. Discussion about the uncertainty in the Lagos study is suggested. Minor revisions as indicated in the comments and remarks below are needed before consideration of publication in ACP.

Detailed Remarks/Suggestions for Revision

Page 4 Line 22: Please define SOs as 'Special Observations (SOs)' here.

Page 5 Line 13: So these levels 'sometimes' could not coincide with the lowest 3 levels? What are the uncertainty introduced here when NH3, HCOOH and CH3OH are calculated based on inconsistent levels of TES products? Can the authors use fixed 3 lowest layers to calculate the values and then compare the results?

Page 8 Line 24 to 34: It is a little bit hard to tell the differences based on the solid line separated by dotted lines. Can the revised manuscript include some values such as 'it contains elevated O3 (xxx ppbv), CH3OH (xxx ppbv) . . .'. Or compile all the satellite measurements (TES, AIRS, and MODIS) in a table?

Page 12 Line 5: The authors emphasize the importance of biomass burning to the local air quality in MCMA and Lagos, what are the possible uncertainties when using GEOS-Chem 2012 simulations driven by the seasonal biomass burning emissions from GFED4? For instance, do 2013-2015 have typical biomass burning scenario as described in GFED4? Further explanation or discussion is suggested.

Page 22: In Figure 1, it is hard to tell the MCMA from the background map. Can the authors use a contour line to highlight the metropolitan area?

Page 29: Same as above, please highlight the Lagos metropolitan area using a circle.

---

## Referee Comment (RC2) · Anonymous Referee #2 · 11 Apr 2017

Overview:

The manuscript by Cady-Pereira evaluates a time series of remote sensing observations of O3, HCOOH, CH3OH and NH3 for several years near Mexico City and Lagos. This unique dataset has the potential to reveal new insights into the trends and distributions of different forms of air pollution in these areas, and to evaluate the extent to which this variability is well represented in models. With these goals in mind, the paper does identify some unique features of the data, such as exploring the impact of biomass burning on an episode of high pollutant concentrations near Mexico City, and evaluating the difference between land-sea gradients in the data vs a model near Lagos. The manuscript is generally clearly written and the topic is suitable for ACP.

[Figure]

However, this paper has several larger shortcomings which would prohibit publication at this time.

The first of these is related to scope. The title and introduction, and even Table 1, seem to lead up to an analysis of megacities world wide. I'm expecting a comprehensive study of satellite data in megacities, such as has been recently published for NO2 and SO2. However, what I find here is a much more narrow look at just 2 cities, despite presenting a Table of 19 cities studied. The reasons for excluding the other 17 cities are never provided. So, this is a bit of a let down when reading through the paper. While it would be tremendously valuable were the authors to extend their analysis to the other 17 cities, I imagine they will resist this suggestion given the effort involved. However, that does mean though that they need to reconsider the scope and aims of the study, and should more succinctly frame the paper in the context of comparing Mexico City and Lagos, and nothing more. What's more, most of the analysis of the data from MCMA is centered around a few biomass burning episodes, which left the authors without much room to consider further analysis of the time series of O3 or CH3OH, which they then state lies beyond the scope of this work.

The second major issue is that the remote sensing products used here don't necessarily reflect the pollutant concentrations at the surface in the urban areas in question, and the extent to which they might will be different for Mexico City vs Lagos. Given the expertise in remote sensing from the authors, this should have been stated and evaluated right up front; rather, it is hardly mentioned, and this just feels like the data is being misrepresented in a way I would have expected more from a group new at satellite data, rather than from the experts. The paper needs to be revised to address this issue head on, and at all stages of analysis throughout the work.

Lastly, the paper tends to read like a bit of a sales pitch for TES. Comparisons of TES to the value from other types of measurements and models is very one-sided. The authors should be more mindful of this throughout.

Details on these comments, as they occurred to me while reading the paper, are described below.

Comments:

1.30: In ascribing these pollutant concentration levels to the cities, is there any concern that the satellite observations are possibly seeing concentrations very different from what is occurring at the surface, or being ascribable to that cities air quality?

2.23 This statement is debatable. NO2 and SO2 gradients near megacities have been well mapped in several studies. Numerous modeling studies provide insight into the key sources and fates of pollution for megacities. I see what the authors are attempting in terms of framing with this sentence, but the wording goes too far.

2.26: Not sure what is meant here by "big picture". My hunch though is it is a very specific interpretation of that phrase that just so happens to be addressed by TES observations. My suggestion though would be to stick to more precise language here, such as the well-made point about vertical distributions.

2.34: This is a pretty one-sided view of satellite observations, where none of the downsides are considered (low spatial resolution, low signal to noise, vertical sensitivities far from the surface, etc.). It seemed odd that the issue of sensing in the free-trop vs at the surface wasn't really considered in the introduction. I'm not even sure setting up a dialogue pitting surface and aircraft observations vs satellite observations is needed — can't the authors just present the science questions and the satellite data, and let the results stand for themselves?

3.24 - 32: This discussion struck me as a bit narrow, not really considering the science questions and literature associated with these species as much as it was brief mention of papers the authors have written studying these species with TES.

4.19-32: The use of SOs is a key component driving this study. As such, I think it should be discussed earlier, in the introduction.

4.29: At this point I'm wondering why the paper is going to focus on just 2 cities rather than these 19.

4.32: I don't know the lat lon of city centers of the top of my head, so it's difficult to evaluate Table 1. Can the authors also include the urban center points, so that we get a sense of the alignment? Or make an array of figures such as that in Fig 1? However, even Fig 1 leaves much to be desired. On what day are these concentration values for? Where is the MCMA region in this picture? More broadly, what is the purpose of considering a true-image color map here? Wouldn't it be more informative to plot the transect over a map that shows the MCMA region and topography (like Fig 2) or to plot over a map of e.g. population? I can't align Fig 2 with Fig 1 since lat / lon aren't specified in the latter, and MCMA isn't shown in the former. Overall, more effective use of maps needs to be considered.

5.31: That is not the correct definition of PM10 (the authors seem to be confusing this with "coarse PM i.e. PM10 - PM2.5). PM10 includes all particles with aerodynamic diameter less than 10. Also, the authors should use the phrase "aerodynamic diameter", not "diameter", in these definitions.

6.30: I'm not sure what is meant by "NH3 emissions are limited". In time? In space? In magnitude or the extent of sectors considered? Please clarify.

7.12: All inventories, or just the EDGAR inventory? For example, this might be very different than inventories constructed specifically for these regions, such as NEI (US) or BRAVO (Mexico).

8.3: Yes, but wouldn't it also be important to say that aircraft studies have linked MCMA pollution to mostly being owing to sources within MCMA? Long-range biomass burning contributions are a small fraction of the air quality problem there. The broader relevance of this episode to Mexico City isn't really made clear.

Fig 4: This isn't a very effective use of space. I think the trajectories and fire locations

could be shown on the same map. The maps may be zoomed over the regions of interest. Other information like the sub-national political boundaries in Guatemala are distracting and should be removed. Overall, I get the sense the authors are using some automated figures generated by different tools rather than synthesizing the data to make their own most effective figures.

8.20: At some altitudes, yes, but for the lower levels the trajectories appear to run north of much of the burning. This would be clearer though if the trajectories and fires were on the same map.

8.29: Fire maps for April 23 not shown?

Fig 6: It would be useful to indicate the latitude of MCMA center and caldera edges in this figure, since they are referred to in the text concerning Fig 6 but not evident here without cross referencing other tables and figures.

9.17: The phrase "air inside the basin was somewhat isolated from the biomass burning influence" summarizes one of my key issues with the presentation of this analysis if the "MCMA air pollution" sources, as to be more precise the analysis appears to be of concentration that are near and high above MCMA but not necessarily indicative of the air pollution at surface level within MCMA itself, and as such the motivation for learning more about them has not been well stated.

9.28: If a critical analysis of the CH3OH trends are beyond the scope of this paper, does inclusion of the data itself warrant being within the scope of this paper? I'm struggling to see the point. At this point in the paper, it seems CH3O3 could be dropped and all of the points made thus far (which are mostly about a biomass burning episode) could be made equally well.

9.27: The authors should know, from the TES averaging kernels, exactly what the TES sensitivity is to near surface O3 concentrations. Hence, this "possible lack" is something that should be rigorously quantifiable rather than speculated.

9.30: Why? This seems like a rather random thing to do. Is one of the goals of this paper really to evaluate the MIROC model accuracy? Is the MIROC model to be used for some analysis to help explain the TES data later on? After reading the entire paragraph it seems the only point is to make the claim that TES can see data at finer scales than larger models. This is a rather obvious point, given the spatial dimension of the TES footprint vs the model resolution, and does little to quantify anything useful for the satellite data or modeling community. One could imagine using an aggregate of satellite data to see if the coarse model gets at least a good estimate of what it is built to estimate, namely average concentrations at the 300 km scale, but that goes way beyond the analysis provided here. As such, I strongly suggest just removing this paragraph entirely, and the associated summary of this point in the abstract.

10.10: If the authors want to use human health impacts as a motivating factor, then they need to more critically discuss the relevance of free-tropospheric concentrations of these species to surface concentrations and health.

10.22: Perhaps, horizontally, but the sensitivity of in situ measurements to concentrations at the surface level would be a benefit for the latter. So, again, this doesn't come across as a balanced assessment, rather than a sales pitch for TES.

10.9 - 11.8: Should the authors decide to limit their paper to just a case study of these two cities, then this content should all be in the introduction.

11.15- 17: It is implied that the TES data reflect Mexico City surface-level concentrations (because that is what has a "reputation"), in which case it would be interesting/shocking that these values are smaller than TES measurements in Lagos, but in fact from the previous analysis we learned that the MCMA TES data doesn't necessarily reflect the Mexico City basin concentrations. In other words, from this comparison I'm just not sure if I've learned anything about the difference of pollution levels between these two cities, or the difference in the ability of nearby TES transects to represent the urban pollution.

Fig 10: What is the date for the NH3 concentrations here? What is the benefit of showing land cover and ocean bathymetry?

11.29: Or because NH3 dry deposits quickly. I'm not sure the evidence presented here alone is sufficient to blame the loss on secondary aerosol formation, although more analysis of how SO2 and NO2 levels vary (or do not) with season might be used to make such distinctions.

Fig 12: What is shown by the vertical dashed line - land sea interface? Why do the JJA and SON CH3OH and HCOOH concentrations plummet to sharply at around 7 degrees?

12.11: TES isn't usually used to evaluate surface-level O3. To what extent is the TES O3 profile at these three lowest levels impacted by the prior compared to the measurement?

Corrections:

1.25: Something is grammatically odd about this sentence, switching from singular "it" to plural "data".

1.25: no comma, or change to "and we show"

3.15: Adjust grammar here: "used used", and "two of TES observing modes"

3.25: subscript 3

3.29: Better worded as "carbon monoxide observed by TES"

4.15: After 2011, SOs...

5.11: In general,

5.23 and elsewhere: use degree symbol rather than "deg" and the multiplication symbol rather than x?

5.30: We "consider" instead of "look at"

6:30: comma after However

7.8: 20%

7.17: http://geos-chem.org

7.25: "if from"?

10.6: double period

11.24, 11.33, other places: O3

11.25: missing space

12.4: per force

---

## Author Comment (AC1) · 17 May 2017

We thank the reviewer for the many comments and suggestions. They have helped make the paper more focused and clearer.

Reviewer comments in blue, our responses in black.

This paper has several larger shortcomings which would prohibit publication at this time.

The first of these is related to scope. The title and introduction, and even Table 1, seem to lead up to an analysis of megacities world wide. I'm expecting a comprehensive study of satellite data in megacities, such as has been recently published for NO2 and SO2. However, what I find here is a much more narrow look at just 2 cities, despite presenting a Table of 19 cities studied. The reasons for excluding the other 17 cities are never provided. So, this is a bit of a let down when reading through the paper. While it would be tremendously valuable were the authors to extend their analysis to the other 17 cities, I imagine they will resist this suggestion given the effort involved. However, that does mean though that they need to reconsider the scope and aims of the study, and should more succinctly frame the paper in the context of comparing Mexico City and Lagos, and nothing more. What's more, most of the analysis of the data from MCMA is centered around a few biomass burning episodes, which left the authors without much room to consider further analysis of the time series of O3 or CH3OH, which they then state lies beyond the scope of this work.

This paper was meant as an introduction to and an illustration of the information in the TES Megacity data. As such we wished to provide some motivation for studying megacities, thus the broad introduction. We agree that the title was misleading and have changed it to: Seasonal and Spatial Changes in Trace Gases over Megacities from AURA TES Observations: Two Case Studies. We have also removed the table, which is now available on the TES website.

The second major issue is that the remote sensing products used here don't necessarily reflect the pollutant concentrations at the surface in the urban areas in question, and the extent to which they might will be different for Mexico City vs Lagos. Given the expertise in remote sensing from the authors, this should have been stated and evaluated right up front; rather, it is hardly mentioned, and this just feels like the data is being misrepresented in a way I would have expected more from a group new at satellite data, rather than from the experts. The paper needs to be revised to address this issue head on, and at all stages of analysis throughout the work.

While we know that TES $O_3$ is not extremely sensitive to the surface $O_3$ levels, there is information from the lower troposphere coming from the retrieval; see new Figure A1 in the Appendix: there is information driving the retrieval away from the a priori, especially for polluted events; thus there is some sensitivity to lower tropospheric $O_3$.

For the other three species ($NH_3$, $CH_3OH$ and $HCOOH$) the physics are favorable for obtaining surface information: all three are concentrated in the boundary layer, and $NH_3$ and $HCOOH$ are radiatively active in spectral windows where there is little absorption from other species. While the TES sensitivity often peaks above the surface for these species, we have found that the surface values are strongly correlated with the measurements at the sensitivity peak. Thus the TES data do provide information on the surface amounts, especially gradients and temporal variability.

We have argued that the levels of the trace gases we studied are markedly higher over the Lagos region than over the MCMA (Section 3.2, Figure 11). This could be attributed to greater sensitivity of the TES retrievals over Lagos, but a comparison of the DOFS (see Figure A2 in Appendix) shows that the sensivity over MCMA overall and over Lagos during the dry season (roughly December to April) are comparable, with the exception of $HCOOH$, which has very elevated concentrations over Lagos in this period, leading to stronger signals.

We have added a short appendix which addresses the issue of information at the surface; we have also added text where relevant in the paper.

Lastly, the paper tends to read like a bit of a sales pitch for TES. Comparisons of TES to the value from other types of measurements and models is very one-sided. The authors should be more mindful of this throughout.

We agree we got carried away by our enthusiasm for the strengths of satellite data. We have reworked comparisons of surface, aircraft and satellite measurements to provide a more balanced view.

1.30: In ascribing these pollutant concentration levels to the cities, is there any concern that the satellite observations are possibly seeing concentrations very different from what is occurring at the surface, or being ascribable to that cities air quality?

Please see above.

2.23 This statement is debatable. NO2 and SO2 gradients near megacities have been well mapped in several studies. Numerous modeling studies provide insight into the key sources and fates of pollution for megacities. I see what the authors are attempting in terms of framing with this sentence, but the wording goes too far.

2.26: Not sure what is meant here by "big picture". My hunch though is it is a very specific interpretation of that phrase that just so happens to be addressed by TES observations. My suggestion though would be to stick to more precise language here, such as the well-made point about vertical distributions.

We have reworked this section and hope it provides a more balanced view of the strengths and weaknesses of each measurement approach.

3.24 - 32: This discussion struck me as a bit narrow, not really considering the science questions and literature associated with these species as much as it was brief mention of papers the authors have written studying these species with TES.

The objective of this section was to provide some brief information to the readers as to why the species are interesting, not to provide a thorough review of the current science questions for each species.

4.19-32 The use of SOs is a key component driving this study. As such, I think it should be discussed earlier, in the introduction.

We have transferred text on the SOs to the introduction, as suggested.

4.29: At this point I'm wondering why the paper is going to focus on just 2 cities rather than these 19.

We had the choice of writing a broad survey of the results of all 19 cities, or carrying out a more in-depth analysis of a few. We chose to follow the latter approach, in order to better illustrate the issues that these data can address, and the kinds of questions that arise.

4.32: I don't know the lat lon of city centers of the top of my head, so it's difficult to evaluate Table 1. Can the authors also include the urban center points, so that we get a sense of the alignment? Or make an array of figures such as that in Fig 1? However, even Fig 1 leaves much to be desired. On what day are these concentration values for? Where is the MCMA region in this picture? More broadly, what is the purpose of considering a true-image color map here? Wouldn't it be more informative to plot the transect over a map that shows the MCMA region and topography (like Fig 2) or to plot over a map of e.g. population? I can't align Fig 2 with Fig 1 since lat / lon aren't specified in the latter, and MCMA isn't shown in the former. Overall, more effective use of maps needs to be considered.

We agree that that our images were not the best choice for demonstrating the location of the cities and the TES pixels. We have created new Figures 1 and 10 to better illustrate the location of the TES pixels over the local terrain. Since Figure 2 was created by the Mexico City Air Quality Department without coordinates, we were not able to add them, but we believe the new Figure 1 clearly shows the outlines of the MCMA, making the two figures easier to compare.  As the caption now states, the TES data shown are the MAM $NH_3$ average.

5.31: That is not the correct definition of PM10 (the authors seem to be confusing this with "coarse PM i.e. PM10 - PM2.5). PM10 includes all particles with aerodynamic diameter
less than 10. Also, the authors should use the phrase "aerodynamic diameter",
not "diameter", in these definitions.

We thank the reviewer for this correction, which has been implemented in the text.

6.30: I'm not sure what is meant by "NH3 emissions are limited". In time? In space? In magnitude or the extent of sectors considered? Please clarify.

The model does not calculate NH$_3$ accurately. We have reworded the text to reflect this.

7.12: All inventories, or just the EDGAR inventory? For example, this might be very different than inventories constructed specifically for these regions, such as NEI (US) or BRAVO (Mexico).

The MIROC model does not use the region specific inventories.  We have added "global" as a qualifier.

8.3: Yes, but wouldn't it also be important to say that aircraft studies have linked MCMA pollution to mostly being owing to sources within MCMA? Long-range biomass burning contributions are a small fraction of the air quality problem there. The broader relevance of this episode to Mexico City isn't really made clear.

We now briefly discuss the results obtained from MOZART-4 in Emmons et al. (2010) .

Fig 4: This isn't a very effective use of space. I think the trajectories and fire locations could be shown on the same map. The maps may be zoomed over the regions of interest. Other information like the sub-national political boundaries in Guatemala are distracting and should be removed. Overall, I get the sense the authors are using some automated figures generated by different tools rather than synthesizing the data to make their own most effective figures.

8.20: At some altitudes, yes, but for the lower levels the trajectories appear to run north of much of the burning. This would be clearer though if the trajectories and fires were on the same map.
8.29: Fire maps for April 23 not shown?

We agree that the original trajectory and fire maps were not user friendly. Following the suggestion of the reviewer we combined the fire locations and back trajectories and added a new figure for April 23.

Fig 6: It would be useful to indicate the latitude of MCMA center and caldera edges in this figure, since they are referred to in the text concerning Fig 6 but not evident here without cross referencing other tables and figures.

The edge of the caldera was already marked. We have added the location of Mexico City center.

9.17: The phrase "air inside the basin was somewhat isolated from the biomass burning influence" summarizes one of my key issues with the presentation of this analysis if the "MCMA air pollution" sources, as to be more precise the analysis appears to be of concentration that are near and high above MCMA but not necessarily indicative of the air pollution at surface level within MCMA itself, and as such the motivation for learning more about them has not been well stated.

We have addressed this issue in section 2.1 and in the appendix. Here is the new text from section 2.1

While these representative values are not direct surface measurements, our experience is the representative values for $NH_3$, $CH_3OH$ and $HCOOH$ will be well correlated with surface values; in other words, the spatial gradients and temporal variability of these values will be very similar to gradients and temporal signals of the surface measurements, (e.g. Pinder et al., 2011, Dammers et al., in preparation). For these species most of the gas is concentrated in or just above the boundary layer, therefore TES is measuring concentrations close to the surface; furthermore, both $NH_3$ and $HCOOH$ are radiatvely active in spectral windows, and will dominate the TES signal in these regions. Moreover, since the TES cross-over time is at 1:30 pm local time, TES is observing at the time of day when the boundary layer tends to be thicker and more well mixed, and thus the TES observation is likely to be closer to the surface value.
Since the $O_3$ averaging kernel peaks at much greater altitudes, $O_3$ is distributed over entire troposphere and its concentration peaks in the stratosphere, the above approach would not provide information on near surface values. The DOFS for the first three levels were analyzed and were found to range between 0.2 and 0.5; thus the retrievals results at these levels have some sensitivity to the $O_3$ amounts in this region, and are not simply being driven by the a priori (see Appendix). Chatfield and Esswein (2012) have shown that $O_3$ partial columns over the first 3 km above the surface have correlations with surface $O_3$ ranging from 0.41 to 0.94 for a set of sondes stations across North America. This altitude range roughly corresponds to the first three TES levels. Based on these two observations we have chosen the average of these first three levels as  a representative value for $O_{3.}$.

9.28: If a critical analysis of the CH3OH trends are beyond the scope of this paper, does inclusion of the data itself warrant being within the scope of this paper? I'm struggling

to see the point. At this point in the paper, it seems CH3O3 could be dropped and all of the points made thus far (which are mostly about a biomass burning episode) could be made equally well.

We have re-evaluated our $O_3$ and $CH_3OH$ data, and decided that we still want to present these results. We believe that difference between May 9 and May 25 is interesting in of itself, and that the difference in spatial variability from NH3 and HCOOH is likely due to a combination of greater influence of transport for O3 and CH3OH and local sources within the city. We have added text to this effect.

9.30: Why? This seems like a rather random thing to do. Is one of the goals of this paper really to evaluate the MIROC model accuracy? Is the MIROC model to be used for some analysis to help explain the TES data later on? After reading the entire paragraph it seems the only point is to make the claim that TES can see data at finer scales than larger models. This is a rather obvious point, given the spatial dimension of the TES footprint vs the model resolution, and does little to quantify anything useful for the satellite data or modeling community. One could imagine using an aggregate of satellite data to see if the coarse model gets at least a good estimate of what it is built to estimate, namely average concentrations at the 300 km scale, but that goes way beyond the analysis provided here. As such, I strongly suggest just removing this paragraph entirely, and the associated summary of this point in the abstract.

Here too we believe we should keep this section, as an additional confirmation of the well know capability of large scale models to replicate "normal" large scale events, and their difficulty in modeling extreme events. For those researchers interested in $CH_3OH$, it also illustrates the information content of TES $CH_3OH$.

10.10: If the authors want to use human health impacts as a motivating factor, then they need to more critically discuss the relevance of free-tropospheric concentrations of these species to surface concentrations and health.

We refer the reviewer to our expanded text in section 2.1 and to the appendix, where we argue that the TES data are correlated to surface amounts, based on the TES sensitivity and time of day of the observations.

10.22: Perhaps, horizontally, but the sensitivity of in situ measurements to concentrations at the surface level would be a benefit for the latter. So, again, this doesn't come across as a balanced assessment, rather than a sales pitch for TES.

Our comment about the value of the TES data here was specific to this region, where there are very few in situ instruments, though it holds for other sparsely monitored areas as well. This is a region for which almost any new data would be a significant increase in the amount of data available, which would not be true of an area like Houston or

LosAngeles, where the TES data are just one component of a much larger data record. We have adjusted the paragraph slightly so as not to downplay the importance of the in situ data.

10.9 - 11.8: Should the authors decide to limit their paper to just a case study of these two cities, then this content should all be in the introduction.

We have debated this point ourselves; as lead author I have decided I prefer the general introduction, followed by specific introductions for each section.

11.15- 17: It is implied that the TES data reflect Mexico City surface-level concentrations (because that is what has a "reputation"), in which case it would be interesting/ shocking that these values are smaller than TES measurements in Lagos, but in fact from the previous analysis we learned that the MCMA TES data doesn't necessarily reflect the Mexico City basin concentrations. In other words, from this comparison I'm just not sure if I've learned anything about the difference of pollution levels between these two cities, or the difference in the ability of nearby TES transects to represent the urban pollution.

We show in the appendix that the TES sensitivity is quite similar in both regions, i.e., for each species the DOFS are at similar levels in the Lagos and MCMA, except for HCOOH. We believe this demonstrates that the observed differences are truly due to different concentrations, and have further argued that these concentrations are correlated with the surface values.

Fig 10: What is the date for the NH3 concentrations here? What is the benefit of showing land cover and ocean bathymetry?

We have replaced figure 10 and now state in the caption that it shows the mean DJF $NH_3$.

11.29: Or because NH3 dry deposits quickly. I'm not sure the evidence presented here alone is sufficient to blame the loss on secondary aerosol formation, although more analysis of how SO2 and NO2 levels vary (or do not) with season might be used to make such distinctions.

We thank the reviewer for pointing out that we neglected to mention deposition, which have now added as a possible mechanism for reduced NH3 over the Gulf.

Fig 12: What is shown by the vertical dashed line - land sea interface? Why do the JJA and SON CH3OH and HCOOH concentrations plummet to sharply at around 7 degrees?

We did neglect to describe the dashed line as the land-sea interface. We have now added

this information to the figure caption. Our hypothesis for the sharp drop in CH3OH and HCOOH is that during the rainy season there is no local production of these species so far north of Lagos, as there is no biomass burning; whatever is produced in the Lagos area gets washed out by the rain before it reaches this northerly region.

12.11: TES isn't usually used to evaluate surface-level O3. To what extent is the TES1G O3 profile at these three lowest levels impacted by the prior compared to the measurement?

Figure A1 in the appendix compares the prior and the measured TES signal in the three lowest levels.

The reviewer suggested the following corrections, which we have implemented unless the text changes rendered them unnecessary, except for the correction at 7.17.

Corrections:
1.25: Something is grammatically odd about this sentence, switching from singular "it" to plural "data".
1.25: no comma, or change to "and we show"
3.15: Adjust grammar here: "used used", and "two of TES observing modes"
3.25: subscript 3
3.29: Better worded as "carbon monoxide observed by TES"
4.15: After 2011, SOs: : :
5.11: In general,
5.23 and elsewhere: use degree symbol rather than "deg" and the multiplication symbol rather than x?
5.30: We "consider" instead of "look at"
6:30: comma after However
7.8: 20%
7.17: http://geos-chem.org: The link in the text is to version used for the runs in the paper.
7.25: "if from"?
10.6: double period
11.24, 11.33, other places: O3
11.25: missing space
12.4: per force

---

## Author Comment (AC2)

We thank the reviewer for the helpful comments and suggestions. They have helped make the paper more focused and clearer.

Reviewer comments in blue, our responses in black.

Page 4 Line 22: Please define SOs as 'Special Observations (SOs)' here.

We have rewritten this section as follows: SOs are more closely spaced (12 to 60 km), and extend over a few hundred to a thousand kilometers, providing more detailed information on regional variability.

Page 5 Line 13: So these levels 'sometimes' could not coincide with the lowest 3 levels? What are the uncertainty introduced here when NH3, HCOOH and CH3OH are calculated based on inconsistent levels of TES products? Can the authors use fixed 3 lowest layers to calculate the values and then compare the results?

It is true that sometimes the lowest three levels may not coincide with those we have selected. Given the fairly limited amount of information in the NH3, CH3OH and HCOOH retrievals, the algorithm tends to adjust the a priori profile where it has the most sensitivity, around the peak of the AK. If the AK peaks significantly above the first three levels, then the mean of the first three is basically the mean of the a priori at those levels. Conversely, if the AK peaks at the first level, then including the third level increases the impact of the a priori on the mean. In effect we are trying to aggregate all the information returned by the retrieval into a single value using a simple method. We have added some discussion on this topic in section 2.1. We hope this is sufficient to address the reviewer's concerns.

Page 8 Line 24 to 34: It is a little bit hard to tell the differences based on the solid line separated by dotted lines. Can the revised manuscript include some values such as 'it contains elevated O3 (xxx ppbv), CH3OH (xxx ppbv) : : :'. Or compile all the satellite measurements (TES, AIRS, and MODIS) in a table?

We agree that this format did not work well in print. We have removed the lines and added filled circles at the dates discussed in the text. We hope this makes the figure more legible.

Page 12 Line 5: The authors emphasize the importance of biomass burning to the local air quality in MCMA and Lagos, what are the possible uncertainties when using GEOS-Chem 2012 simulations driven by the seasonal biomass burning emissions from GFED4? For instance, do 2013-2015 have typical biomass burning scenario as described in GFED4? Further explanation or discussion is suggested.

We have added plots of MODIS AOD and AIRS CO for 2012 in the appendix; based on these variables the variability in 2012 was fairly similar to 2013-2015, though AOD reached the high levels observed in 2015. We have also added some more discussion in

the text. The goal of this model comparison was not validate the model but to demonstrate the additional information that can be obtained from the TES data.

Page 22: In Figure 1, it is hard to tell the MCMA from the background map. Can the authors use a contour line to highlight the metropolitan area?

Page 29: Same as above, please highlight the Lagos metropolitan area using a circle.

We have completely revised the transect plots (Figure 1 and Figure 10); we hope the reviewer will find them much more informative.

---

## Author Comment (AC3)

[revised manuscript text omitted]

Karen Cady-Pereira 5/17/2017 10:57

Karen Cady-Pereira 5/17/2017 10:57

Karen Cady-Pereira 5/17/2017 10:57

Karen Cady-Pereira 5/17/2017 10:57

Karen Cady-Pereira 5/17/2017 10:57

Karen Cady-Pereira 5/17/2017 10:57

TES measurement on April 23 (blue circle in Figure 3) is interesting in that it contains elevated $O_3$, $CH_3OH$ and HCOOH, all of which could have a fire source, but low $NH_3$. The fire map and back trajectory for this day (Figure 4, bottom panel) show winds from the same direction as on May 9, but somewhat fewer fires;, the low values of $NH_3$, which has a shorter life than the other species, could indicate that these fires were further away or weaker than on May 9. By June 10 (brown circle in Figure 3) the TES transect (the second point after the May 9 peak), shows low values for all species, indicating cleaner air.

Surface measurements of precipitation, $PM_{2.5}$, $PM_{10}$ and CO (Figure 5) show that the late April-early June period in 2013 can be divided into two regimes: a dry regime from April 27 through May 11, with higher concentrations of $PM_{2.5}$, $PM_{10}$ and CO, peaking around May 9, followed by a wet regime through June 2, with lower pollutant concentrations and a number of rainy days. Tzompa-Sosa et al. (2016) calculated LEV/WSOC ratios during May 2013, which provide an estimate of the contribution of biomass burning to the total WSOC in an air mass. Their results show higher LEV/WSOC ratios during the first weeks of May, compared to the rest of the month. Taken altogether, the data indicate that biomass burning contributes much more to pollution on May 9 than May 25, due to a combination of transport and the lack of precipitation to drive wet deposition. It is, in fact, likely that the changing meteorological conditions associated with the wetter weather were also responsible for the shift in transport away from the fire locations. Note that on June 10[th], when TES data suggest cleaner air, the in situ measurements show rain and lower values of CO and particulate matter.

However, it is important to note the impact of biomass burning varies spatially. Figure 6 shows that $NH_3$ and HCOOH concentrations are extremely high to the south of the Mexico City basin on May 9 and drop off markedly as TES moves north of 19.1° and elevation increases; on May 25 the decrease in $NH_3$ and HCOOH with elevation is much more gradual, and there is actually an increase as the TES observes the air mass inside the basin. The overall abundances of $NH_3$ and HCOOH are higher on May 9 than May 25, which is consistent with the in situ observations, but the difference between the days is much more marked south of the caldera. These results suggest that on May 9 the air inside the basin was somewhat isolated from the biomass burning influence due to the fact that the air from the fires had to travel from the southwest up and over the mountains to reach MCMA (Figure 4, top panel). Seasonal means of $NH_3$ and HCOOH (Figure 7) along the transect path show that this a regular feature, especially in the December through May period: concentrations decrease sharply as the transect passes north over the highest elevation point, at 3088 m, just south of Mexico City, though the MAM HCOOH mean reaches its minimum slightly further north. This result is in agreement with the Emmons et al. (2010) model output, which shows only a a weak influence of biomass burning on the average pollution within the city.

Both $O_3$ and $CH_3OH$ are generally higher on May 9 than on May 25, confirming the high pollution level of this day as indicated by $NH_3$ and HCOOH. On the other hand, the spatial variability along the

Karen Cady-Pereira 5/17/2017 10:57

Karen Cady-Pereira 5/17/2017 10:57

Karen Cady-Pereira 5/17/2017 10:57

Karen Cady-Pereira 5/17/2017 10:57

Karen Cady-Pereira 5/17/2017 10:57

Karen Cady-Pereira 5/17/2017 10:57

[revised manuscript text omitted]

---

## Author Response (AR2)

**Responses to second review by reviewer 2 on "Seasonal and Spatial Changes in Trace Gases over Megacities from AURA TES Observations: Two Case Studies"**

Reviewer comments in black, author comments in blue

The revised manuscript by Cady-Pereira et al. well address most of the reviewer comments. The figures
are significantly improved, and there is additional discussion about the relationship between remote sensing measurements and surface concentrations. I only have a few remaining requests, which are either further clarifications or reiterating a point made in the previous review which the authors chose not to address. These constitute minor revisions.

New comments:

The "Lagos" measurements aren't really over Lagos — they're all to the west, over what seems to be rural land areas. The authors need to be careful that their results are interpreted as being indicative of
urban Lagos concentrations; otherwise, it's not clear how this fits within the framing of the "megacities" aspect of the paper.

It's not clear which pixels across the entire transects (e.g., Fig 12) are used to represent "Lagos" in Fig 11. This is in contrast to MCMA, where Fig 6 shows the portion of the transect (dashed vertical lines)
that is used to represent the city. For example, NH3 levels in the Lagos transects peak at latitudes north of 7 N, but elsewhere in the manuscript it is indicated that the most urban impacted area of the transect is 6.4 N - 7 N (page 13, lines 11-14).

We agree that this required clarification. We have modified the text on the Lagos transects to read:
**TES transects were carried out slightly to the west of the Lagos area (Figure 10); the southern end of the transect, south of 6.2°N, was over the Gulf of Guinea. The elevation is low and constant; there is no complicated topography affecting winds and transport as there is in the MCMA, thus we expect that emissions from the city will strongly impact the surrounding areas, and that the TES measurements will provide information on pollution levels in the greater Lagos**
**metropolitan area. Means of the TES observations along the entire extent of the transect (Figure 11, left panel, solid line) show a strong seasonal pattern in all four species, with clear enhancements during the December-March period, which correlate with the AIRS CO and MODIS AOD seasonal variability (Figure 11, right panel).**

13.8: extra space

Fixed

Revisiting previous comments: mine (with original page/line numbers), their response (A), and my additional response (R)

3.24 - 32: This discussion struck me as a bit narrow, not really considering the science questions and literature associated with these species as much as it was brief mention of papers the authors have written studying these species with TES.

(A) The objective of this section was to provide some brief information to the readers as to why the species are interesting, not to provide a thorough review of the current science questions for each species.

R: Yes, but surely some of the brief information regarding why these species are interesting might be present in articles from outside the TES group? Please expand the set of literature considered here.

We have added a number of references we believe will provide the interested reader more background on the science issues for each species.

9.30: Why? This seems like a rather random thing to do. Is one of the goals ofthis paper really to evaluate the MIROC model accuracy? Is the MIROC model to be used for some analysis to help explain the TES data later on? After reading the entire paragraph it seems the only point is to make the claim that TES can see data at finer scales than larger models. This is a rather obvious point, given the spatial dimensionof the TES footprint vs the model resolution, and does little to quantify anything useful for the satellite data or modeling community. One could imagine using an aggregateof satellite data to see if the coarse model gets at least a good estimate of what it is built to estimate, namely average concentrations at the 300 km scale, but that goesway beyond the analysis provided here. As such, I strongly suggest just removing this paragraph entirely, and the associated summary of this point in the abstract.

(A) Here too we believe we should keep this section, as an additional confirmation of the well know capability of large scale models to replicate "normal" large scale events, and their difficulty in modeling extreme events. For those researchers interested in CH3OH, it also illustrates the information content of

TES CH3OH.

R: I still disagree. The authors admit this is "additional confirmation" of something that is "well known". So, I don't see what it adds.

We leave it to the editor to decide whether this section should be kept or not, as the reviewer and the authors are still in disagreement.

**Responses to editor**

We have add the wavelength information on the MODIS AOD; we have also fixed the x-axis on Figure A3.

[revised manuscript text omitted]

Karen Cady-Pereira 7/3/2017 10:24

Karen Cady-Pereira 7/3/2017 10:24

Karen Cady-Pereira 7/3/2017 10:11